

# Multidecadal variability of the ITCZ from the Last Millennium Extreme Precipitation Changes in Northeastern Brazil

Isela L. Vásquez P.[1], Humberto Alves Barbosa[2], Gilvan Sampaio[1], Cesar Arturo Sánchez P.[3], Giselle Utida[4], David Pareja Quispe[5], Juan G. Rejas Ayuga[6,7], Hugo Abi Karam[8], Jelena Maksic[1], Marilia Harumi Shimizu[1], and Francisco William Cruz[4]

[1]Weather Forecast and Climate Studies Center ,CPTEC. National Institute for Space Research (INPE), Brazil
[2]Federal University of Alagoas (UFAL), Brazil
[3]Applied Computing (CAP), National Institute for Space Research (INPE), Brazil
[4]Institute of Geosciences, University of São Paulo (USP), São Paulo, 05508, Brazil
[5]Department of Interdisciplinary Physics, National University of San Marcos (UNMSM), Peru
[6]Department of Space Programs, National Institute for Aerospace Technology (INTA), Spain
[7]Technical University of Madrid (UPM), Spain
[8]Geoscience Institute, IGEO, Federal University of Rio de Janeiro (UFRJ), Brazil

**Correspondence:** Isela L., Vásquez P. (iselavp@gmail.com)

**Abstract.** Decadal and multidecadal variability of the Intertropical Convergence Zone (ITCZ) is analyzed in space-time using CMIP6 simulations and paleoprecipitation records during the Last Millennium. We investigated the persistence patterns of the CMIP6 ensemble models, using low frequency component analysis (LFCA) to isolate the mechanisms that modulate the ITCZ at the multidecadal scale. The results suggest that the north-south displacement of the ITCZ was related to the oceanic region

with the highest sea surface temperature (SST) of the tropical South Atlantic basin. The zonal mode variability is initially associated with the equatorial region (between 5°S and 5°N) and with the northwestern African coast. These observations also contrast with the paleoclimatic records of the region, indicating a northward shift of the ITCZ during the MCA and a southward shift during the LIA. Based on the periodicities observed the 21 years is predominant during the Last Millennium can be associated with the solar cycle influence on the pattern of ITCZ contracted and positioned in the central region of

the equator. This relationship suggests that, although ENSO is the main driver in variability over Tropical South America at interannual time scales, this influence can be significantly modulated by longer time scales. The results suggest the existence of a low-frequency variability, modifying the distribution of precipitation and with consequences in the intensity and frequency of droughts/floods events in the NE, indicating that these events are associated with the coupling between the oceans and the atmosphere.

**Keywords.** Tropical Atlantic. Intertropical Convergence Zone. Little Ice Age. Medieval Climate Anomaly. CMIP5. CMIP6.



## 1 Introduction

The Intertropical Convergence Zone (ITCZ) is a narrow band located near the equator where the northerly and southerly trade winds meet and converge creating clusters of clouds and precipitation maxima (Waliser and Gautier, 1993; Philander et al.,

1996). The ITCZ position, intensity, and dynamic are the result of ocean/atmosphere and ocean/land coupling (Marshall et al., 2014). Seasonal excursions of the ITCZ, to the north during the boreal autumn/summer, and to the south during the austral autumn/summer, are controlled by seasonal cycle of solar insolation. These seasonal shifts vary with longitude, where over the Atlantic and eastern Pacific oceans the ITCZ resides north of the equator, while over the Indian ocean it is located mostly south of the equator (Liu et al., 2020). As the Atlantic sector of ITCZ has a strong influence on tropical precipitation and climate

of northern Brazil thus affecting many societies and ecosystems, it is important to comprehend it on different time scales. In the present-day climate, the annual mean position of the Atlantic ITCZ is around 5°N, with large interannual variability. Observations suggest that the interannual variability of Atlantic ITCZ position is determined by both sea surface temperature (SST) of tropical Atlantic (Nobre and Shukla, 1996), and by tropical Pacific Ocean (Giannini et al., 2001; Chiang et al., 2002; Tedeschi et al., 2013). On a time scale longer than interannual the ITCZ position is modulated by both tropical and extratropical

factors, and their complex interactions. Numerous studies are suggesting that changes in the Atlantic meridional overturning circulation (AMOC) and Atlantic multidecadal variability (AMV) have significant impact on the ITCZ position (Knight et al., 2006; Buckley and Marshall, 2016; Green et al., 2017; Zhang et al., 2019). Green et al. (2017) also found correlations between decadal and multidecadal variability in the midlatitude North Pacific and the ITCZ shift.

Valuable insights into ITCZ behavior are gained with the use of paleorecords, proxies for tropical hydrologic changes, in

recent periods before climate warming. During the "Medieval Climate Anomaly" period (MCA, 900 – 1150 AD, hereafter all dates are AD). Periods of intense summers and major droughts were recorded in several regions of the Northern Hemisphere (Kleppe et al., 2011; Vuille et al., 2012). The arid regions of the middle latitudes of the northern hemisphere became more humid during the "Little Ice Age" period (LIA, 1500 – 1850).

According to decreased detrital delivery from local rivers to Cariaco Basin, the mean latitudinal position of ITCZ shifted

southward during the LIA (Haug et al., 2001; Peterson and Haug, 2006). While Haug et al. (2001) point to Pacific-based climate variability as a driver for this shift, Peterson and Haug (2006) suggest that it could be a response to forcing originating in the high latitude of the Atlantic Ocean. Several other high-quality records from the Neotropics reaffirm persistent southward displacement of the ITCZ during the LIA (Vuille et al., 2012; Lechleitner et al., 2017). However, recent paleoclimatic studies suggest the expansion and contraction of the ITCZ during the last millennium (Utida et al., 2019; Asmerom et al., 2020).

Due to scarcity of paleorecords and complex relationship between ITCZ position, width and intensity (Byrne et al., 2018), understanding the mechanisms that regulate the position of the ITCZ is still limited.

Use of climate models provide dynamical/thermodynamical variables that can improve the comprehension of these processes and fill an important knowledge gap. To contribute to Coupled Model Intercomparison Project Phase 5 (Taylor et al., 2012) and Phase 6 (Eyring et al., 2016) Last Millennium experiments (period 850 – 1850) have been incorporated in the Paleoclimate

Modeling Intercomparison Project. According to Ortega et al. (2021), biases in representing the ITCZ during summer and



spring seasons are reduced in the CMIP6 models with respect to CMIP5. However, the double-ITCZ bias over the Atlantic is only slightly reduced and big inter-model spread still remains in the CMIP6 models (Tian and Dong, 2020).

## 2 Study area and data description

### 2.1 Study area

The variability of the SST in the Tropical Atlantic (TA) is important for the population of Africa and South America as it influences the precipitation regime (Seager et al., 2010). Specifically in NEB, the main physical variables influencing climate variability conditions are the SST of the TA and tropical Pacific oceans (Philander, 1989).

In the Equatorial Atlantic (EA), the southern gradients of SST have a profound impact on the total precipitation in the NEB through the modulation of the latitudinal positioning of the ITCZ (Hastenrath and Greischar, 1993; Hastenrath, 1984; Nobre

and Shukla, 1996; Marengo, 2004; Chang et al., 2006; Marengo et al., 2017; Deser et al., 2010). In response to the warming of the SST in the Atlantic and the variation in the intensity of the trade winds, the ITCZ presents a latitudinal displacement throughout the year (Waliser and Gautier, 1993; Waliser and Jiang, 2015). Between February and March, it is located between the equator and 5°N, due to higher SST in the South Atlantic during the austral summer and the intensification of northeast trades. Between July and August, it is located approximately between 5°N and 8°N in the western part of the TA, a position

resulting from the highest SST in the North Atlantic and the rise of the high pressure centers in the Azores and Santa Helena, which causes intensification of southeast trades and weakening of northeast trades (Peterson and Stramma, 1991; Wagner, 1996; Hastenrath, 2006). Seasonal migration of the ITCZ is the main mechanism that induces precipitation in the TA region (Hastenrath and Greischar, 1993) and variations in its migration are related to SST anomalies in the TA, known as the Atlantic Meridional Mode (AMM). In northeastern Brazil (NEB), ITCZ is considered dominant in determining precipitation (Hastenrath

and Heller, 1977; Hastenrath, 1997; Hastenrath and Lamb, 1977).

### 2.2 Description of the dataset

The model evaluation was based on historical simulations of CMIP5 and CMIP6 models shown in Table 1. The main objective is a quantitative assessment in the estimation of the variability of the ITCZ in the tropical Atlantic. A first analysis was to compare the average annual rainfall cycles simulated by the CMIP5 and CMIP6 models in comparison with the observed

pattern (GPCP).

Last Millennium (LM) experiments to contribute Coupled Model Intercomparison Project Phase 6 (Eyring et al., 2016) have been incorporated in Paleoclimate Modelling Intercomparison Project Phase 4 (PMIP4). These simulations cover the period (850 – 1850). Models can be downloaded from: https://esgf-node.llnl.gov/search/cmip6. We used the LM outputs from 2 Models. They were MIROC–ES2L (Hajima et al., 2019) and MRI–ESM2–0 (Yukimoto et al., 2019). These models were

selected because they were the only ones that contained, simultaneously, data from the five climatic variables considered for: from the last millennium (850 – 1850). We only selected the first and second realization because of the lack of the first



**Table 1.** List of outputs from the CMIP5 and CMIP6 models used in this study.

| N° | Institution | CMIP5 | CMIP6 |
|---|---|---|---|
| 1 | Meteorological Research Institute (MRI) | MRI-ESM1 | MRI-ESM2-0 |
| 2 | Atmosphere and Ocean Research Institute (AORI) | MIROC-ESM | MIROC-ES2L |
| 3 | National Aeronautics and Space Administration (NASA) | GISS-E2-H | GISS-E2-1-G |
| 4 | Met Office Hadley Centre | HadCM3 | HadGEM3-GC31-LL |
| 5 | Institut Pierre-Simon Laplace (IPSL) | IPSL-CM5A-LR | IPSL-CM6A-LR |
| 6 | Max Planck Institute for Meteorology (MPI-M) | MPI-ESM-P | MPI-ESM1-2-LR |
| 7 | Beijing Climate Center (BCC) and China Meteorological Administration (CMA) | BCC-CSM-1-1-m | BCC-CSM2-MR |
| 8 | Atmosphere and Ocean Research Institute (AORI) | MIROC5 | MIROC6 |
| 9 | Australian Community Climate and Earth System Simulator (ACCESS) | ACCESS1-0 | ACCESS-CM2 |

realization in these models (r1i1p1f2 and r1i1p1f1, respectively). The variables monthly scalar used are: precipitation, sea-level pressure (SLP), sea surface temperature (SST), the zonal and meridional wind components (u, v), the Lagrangian tendency of air pressure ($\omega$) are used in the present study.

For investigate the seasonal of the ITCZ, we adopted Global Precipitation Climatology Project (GPCP) monthly data with a spatial resolution of 2.5°x2.5° (Latitude and Longitude) (Adler et al., 2003) and Outgoing longwave radiation (OLR) monthly data, collected of the National Oceanic and Atmospheric Administration (NOAA). These data have 2.5° x 2.5° grids (Latitude by Longitude) and were used for the period 1981-2005. The analysis is based on the inverse relationship between the OLR. (Liebmann and Smith, 1996) observed rainfall: High OLR is associated with a lower chance of rainfall and low OLR with a

higher chance.

We compared the simulations using multiproxy reconstructed data paleoclimatic records from the South American. In particular, this study will take advantage of proxy precipitation data published for (Vuille et al., 2012; Utida et al., 2019). The seasonal climate in the coastal regions of the tropical Atlantic as in the NEB and in the Cariaco basin (Coordinates: 10°42' N, 65°10' W) are mainly controlled by the displacement of the ITCZ (Utida et al., 2019; Campos et al., 2019). Dam Boqueirão

is located close to the east coast, in the NEB region, where $\sim$ 50% of the precipitation originates during the ITCZ's southern position in the MAM months (March - April - May) is associated with warmer sea surface temperatures (SST) in the Tropical South Atlantic (TSA) during the austral winter JJA (June - July - August) is characterized by significant cooling in the TSA, stronger southeast trade winds that cross the Equator and a northward displacement of the ITCZ (Utida et al., 2019). The record of the hydrogen isotope composition of the n-C28 alkanoic acid ($\delta$Dwax) from the core Boqc0901 dated (Viana et al., 2014).

Changes in lower (higher) $\delta$Dwax reflect changes of ITCZ related precipitation (Zocatelli et al., 2012; Viana et al., 2014; Utida et al., 2019). The Cariaco record shows the ratio of Titanium in the Cariaco Basin (Haug et al., 2001) to decrease in titanium (Ti) concentration in Cariaco Basin sediment suggests reduced rainfall (increased aridity). Ti peaks in the Cariaco sedimentary



record, suggesting that ITCZ has been shifted to the northern position (Haug et al., 2001). In order to reveal possible association with AMO variability, the index AMO is calculated detrend that time series (Enfield et al., 2001) from the area-weighted North Atlantic (0° - 70°N and 80° - 0°W).

## 3 Methods

### 3.1 Low-frequency component analysis (LFCA)

LFCA is a method that transforms the leading empirical orthogonal functions (EOFs) of a data set in order to identify a pattern with the maximum ratio of low-frequency to total variance (based on application of a low-pass filter). The resulting low-frequency patterns (LFPs) and low-frequency components (LFCs). This method is presented in (Wills et al., 2018). To analyze the variability of the ITCZ associated with LIA and MCA, we use LFCA of SST and precipitation anomalies (between 30°S and 30°N) and SST in the AT region with the highest ratio of low-frequency to total variance in long the last millennium simulations monthly the SST and precipitation of the ensemble (Averaging the models MRI-ESM2-0 and MIROC-ES2L).

### 3.2 Hodrick-Prescott smoothing (HP)

HP is a linear filter that requires previous specification of the parameter, which in a joint analysis should be the same for the filtering of all series; the higher its value, the resulting series will be smoother (Hodrick and Prescott, 1997). For the monthly time series, the value 14400 is the most frequently used (Maravall and Del Rio, 2001).This value filters out from the time series tendency possibly existing less than 10 years and interdecadal cycles (Vásquez et al., 2018). In order to reveal possible trends, phase synchrony and the temporal variability of precipitation in the data paleoclimatic records and the components LFCs precipitation CMIP6 models and their possible association with AMO variability, were applied to these time series: Hodrick-Prescott smoothing (the so-called "HP filter").

### 3.3 Probabilistic tracking of the ITCZ

We estimated probability density functions (PDF) for rainfall as proposed by Mamalakis and Foufoula-Georgiou (2018), this method facilitates the detailed analysis of changes in the seasonal dynamics of the ITCZ and offers the possibility of using multiple variables to define it, which increases its physical rigor. Furthermore, it is computationally efficient and flexible in its implementation, which makes it useful for the analysis of multi-model ensembles in climate change assessment studies. In this study, PDF analysis was applied to determine the climatology of the ITCZ, particularly its seasonal and annual average location as a function of explicit longitude, with the objective of evaluating the relative capacity of the CMIP5 and CMIP6 Models (Table 1) to reproduce the inter-annual and intra-seasonal variability of precipitation.



### 3.4 Red-noise model and statistical significance

The univariate spectra were bias-corrected using 1000 Monte Carlo simulations. This was carried out by use of the publicly available REDFIT, which automatically produces first-order autoregressive (AR1) time series with sampling times and characteristic timescales matching those of the real climate data. This approach has been widely used in the components LFCs precipitation CMIP6 models and their possible association with AMO variability, according to their age, given by the midpoint of the time window, and coloured-coded the frequencies for which the spectral power exceeded red-noise false-alarm levels (confidence levels) of 95%.

## 4 Results and discussion

### 4.1 Probabilistic The location of the ITCZ

To identify the uncertainties in the simulations by the CMIP5 and CMIP6 models, in terms of the position of the ITCZ we used a probabilistic method, obtaining a spatial average of the probability distribution of the seasonal location of the ITCZ over the tropical Atlantic (Fig. 1). They are described below according to seasonality in the Southern Hemisphere: from December to February (DJF - summer), from March to May (MAM - autumn), from June to August (JJA - winter) and from September to November (SON - spring).

Our results indicate that the CMIPs models simulate a migration of the ITCZ towards the south in relation to those observed in the DJF and MAM periods. These results show that the Atlantic bias has a greater magnitude relative to the JJA and SON periods. This implies that the contraction pattern of the ITCZ over the Atlantic Ocean (Fig. 1) is likely a result originating from some of the models that are heavily biased during the base period. This suggests that the presence of coupling and a double-ITCZ are important factors to allow the transition of the polarization regime. Furthermore, there is a clear underestimation of the simulated precipitation in the ITCZ band. It is also worth mentioning that the models have a clear tendency to reproduce a double band of the ITCZ in the Atlantic Ocean, a fact that is reflected in causing errors in the simulation of accumulated precipitation, nevertheless, the models manage to simulate the southward (north) displacement of the ITCZ during the summer (winter), corroborating the results obtained in other works (Wang et al., 2010; Carvalho and Cavalcanti, 2016; Wang et al., 2017; Mamalakis et al., 2021). According to Uvo (1989), this great variability may be associated with different large-scale transient systems operating in the South American and African continents that affect the ITCZ. The difference in the location of the ITCZ observed in the different climate models in the different seasons in relation to the current climatology of the ITCZ is due to the presence of biases in the model (Mamalakis et al., 2021). In general, these biases have patterns that are largely season-independent (Fig. 1), and translate into a decrease on annual timescales (Fig. 2).

With respect to interannual variability, the CMIP models tend to simulate a double-ITCZ during the historical period. Thus, according to the CMIP simulations in the LIA period (MCA) simulates a strong band of the ITCZ shifted to the Southern Hemisphere (Fig. 3), there was not a marked difference in the position of the ITCZ. According to (Rojas et al., 2016), this southward displacement of the ITCZ, during the austral summer in the LIA period, was weak and insignificant for the simula-





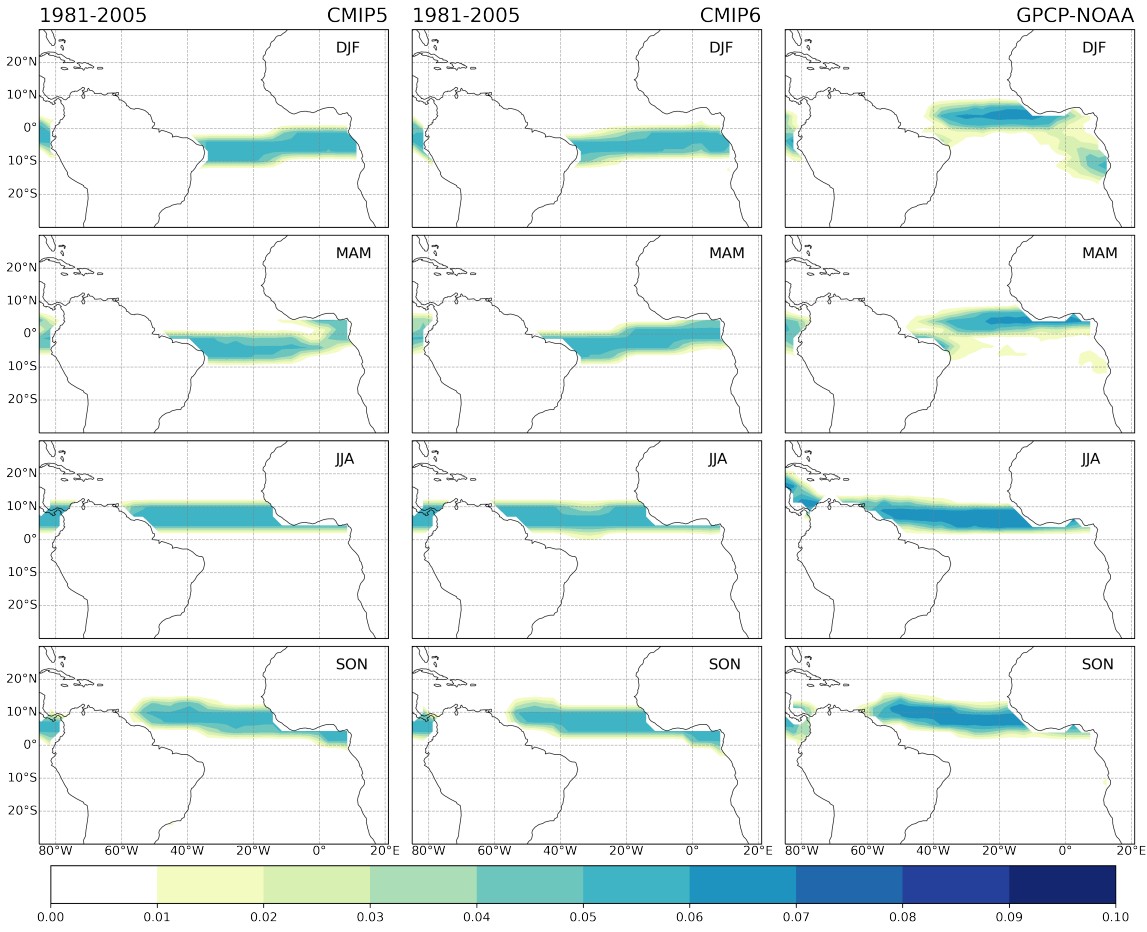

**Figure 1.** Location of the ITCZ during 1981-2005 based on observations and CMIP6 models. a) Probability density function (PDF) of the location of the ITCZ in all longitudes according to the seasonality in the South hemisphere: The ITCZ tracking is performed based on the joint statistics of the observed (panel (a)) or the simulated (panel (b)) window-mean precipitation (from GPCP) and outgoing longwave radiation OLR (from NOAA); in overlapping longitudinal windows. the multi-model mean across all 9 CMIP6 models is presented.

tions of the PMIP3 models, which suggests that the main factor influencing precipitation in NE Brazil is the overestimation of ITCZ precipitation (Yin et al., 2013; Rojas et al., 2016).

Mamalakis et al. (2021) observed an opposite response from the ITCZ driven by a positive SST pattern. This opposite ITCZ response can be summarized as a northward shift over East Africa and the Indian Ocean and a southward shift over the eastern Pacific, South America and the Atlantic.Our results indicate that the CMIP5 and CMIP6 models generally simulate a more frequent southward migration of the ITCZ than is observed, the seasonal bias of double ITCZ on the annual scale is evident mainly over the South Atlantic (Fig. 2).This finding is in contrast to recent studies (Richter and Xie 2008; Richter et al. 2013; Zermeno-Diaz and Zhang 2013, Shonk et al., 2019; Mamalakis et al.2021).




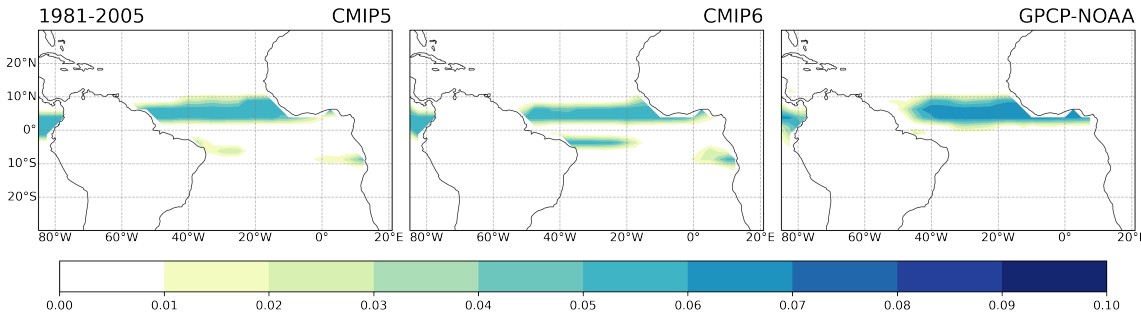

**Figure 2.** Probability density function (PDF) of the location of the ITCZ on annual scales and in all longitudes. The ITCZ tracking is performed based on the joint statistics of the observed window-mean precipitation and outgoing longwave radiation (OLR); in overlapping longitudinal windows. the multi-model mean across all 9 CMIP6 models is presented.

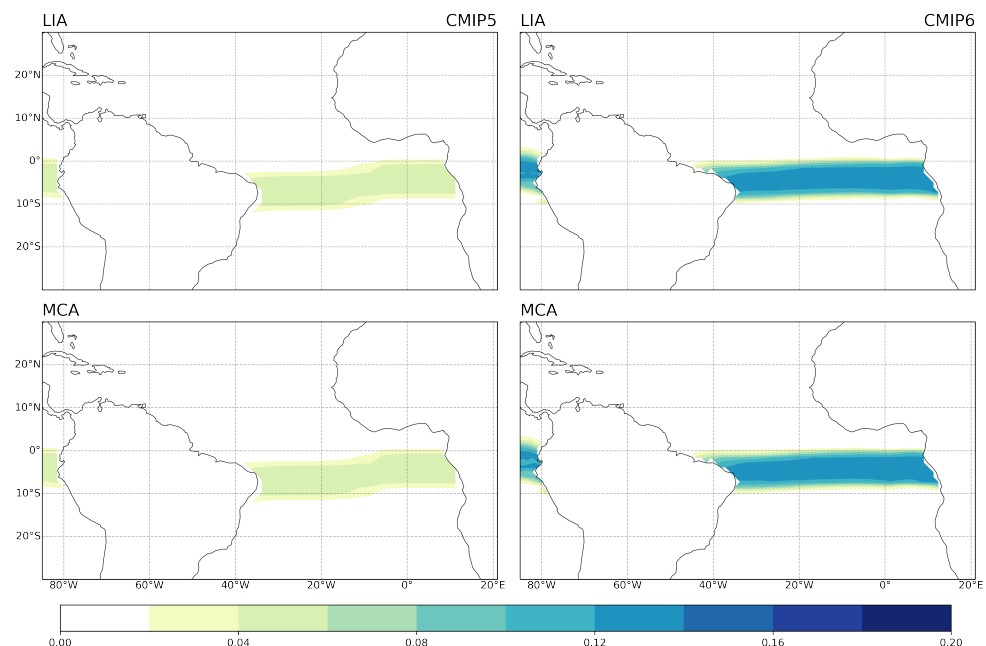

**Figure 3.** Probability density function (PDF) of the location of the ITCZ on annual scales and in all longitudes. The ITCZ tracking is performed based on the joint statistics of the observed window-mean precipitation and outgoing longwave radiation (OLR); in overlapping longitudinal windows. Last millennium in the ensemble CMIP5 (MRI-ESM1 and MIROC5) and CMIP6 (MRI-ESM2-0 and MIROC-ES2L).

Recently Huang et al. (2013) evaluated the contribution of extratropical biases to the double-ITCZ tropical in CMIP5 models. They revealed robust statistical relationships between the double-ITCZ tropical bias, northward atmospheric energy transport, and radiative forcing from extratropical shortwave clouds. The authors argued that cloud bias over the Antarctic ocean can result in too much shortwave radiation and warm bias in the mid-latitudes of the Southern Hemisphere.





Hwang and Frierson (2013) found that the double ITCZ problem in the models is due to the bias of the energy balance
between the two hemispheres. They observed that excessive energy absorbed in the Southern Hemisphere was transported to
the Northern Hemisphere via the upper branch of the Hadley anomaly cell over the equator, while the lower branch transported
water vapor southward and caused a southward shift of the ITCZ, inducing the formation of the double ITCZ. Other studies
attribute the displacement of the ITCZ to the south, due to the rapid increase in anthropogenic emissions of aerosols (Wang,
2015; Chung and Soden, 2017), through various physical mechanisms (Zhang et al., 2021).

**4.2   The Interdecadal Component temporal of the ITCZ latitudinal location**

This study isolates the variability multidecadal on precipitation and of the Atlantic sea surface temperature by using low
frequency component analysis (LFCA). Our analysis identifies the multidecadal Atlantic SST variability over the subpolar
North Atlantic. To investigate latitudinal ITCZ displacements, we compare the data from Boqueirão Lake to the Ti record
from Cariaco Basin; these southern variations of the ITCZ could be marked by the antiphasic relationship with the Cariaco
Basin. There was evidence of migration from the ITCZ further south in relation to its current position. In general, the results
of the LFCA analysis indicate that the SST and precipitation variables of the ensemble (Averaging the models MRI-ESM2-0
and MIROC-ES2L) in the last millennium are capable of reproducing the large-scale changes in the location of the ITCZ in
the Atlantic and the precipitation during the LIA and MCA periods, these results are consistent with the evidence from the
paleoclimatological records.

When comparing the trends in the $\delta$Dwax data with the LFC1 of precipitation, we observed that along the MCA, the
models registered a positive trend (dry periods) and a negative trend (wet periods), with negative trends predominating. When
comparing the trends in the $\delta$Dwax data with the LFC2 of precipitation during the MCA, it is observed that for the ensemble
model, negative trends predominate over the period, different from the ensemble model where positive trends predominate.
When comparing the trends in the $\delta$Dwax data with the LFC1 and LFC3 of precipitation, we observed that both models show
predominant positive trends (wet periods) along the LIA (Fig. 4).

The trends in %Ti records from the Cariaco Basin during MCA (LIA) showed wet (dry) conditions in the Cariaco basin
(Fig. 2). We analyzed the LFC2 and LFC4 of the models and observed that throughout the MCA, positive trends predominate,
in agreement with what was observed in this period (Fig. 5.a). During the LIA, the trends of the LFCs show a high variability.
Note the LFC4 of the ensemble model, the positive trends predominate. In this period the ITCZ's position is further south,
indicating the occurrence of drought (Haug et al., 2001; Novello et al., 2012; Utida et al., 2019). When comparing the trends of
the Ti records in the Cariaco basin with those of the DV2 sample in the MCA-related period, the records describe an antiphasic
behavior, indicating that the ITCZ's northernmost position.

It is generally observed that, during the study period, the components of the trends of the LFCs have a high variability, being
accentuated during the LIA. The MCA and LIA comparisons between the model and the proxies are not well represented in all
components. More proxies data from the region between Cariaco and Boqueirão are needed to better evaluate the representation
of the models.





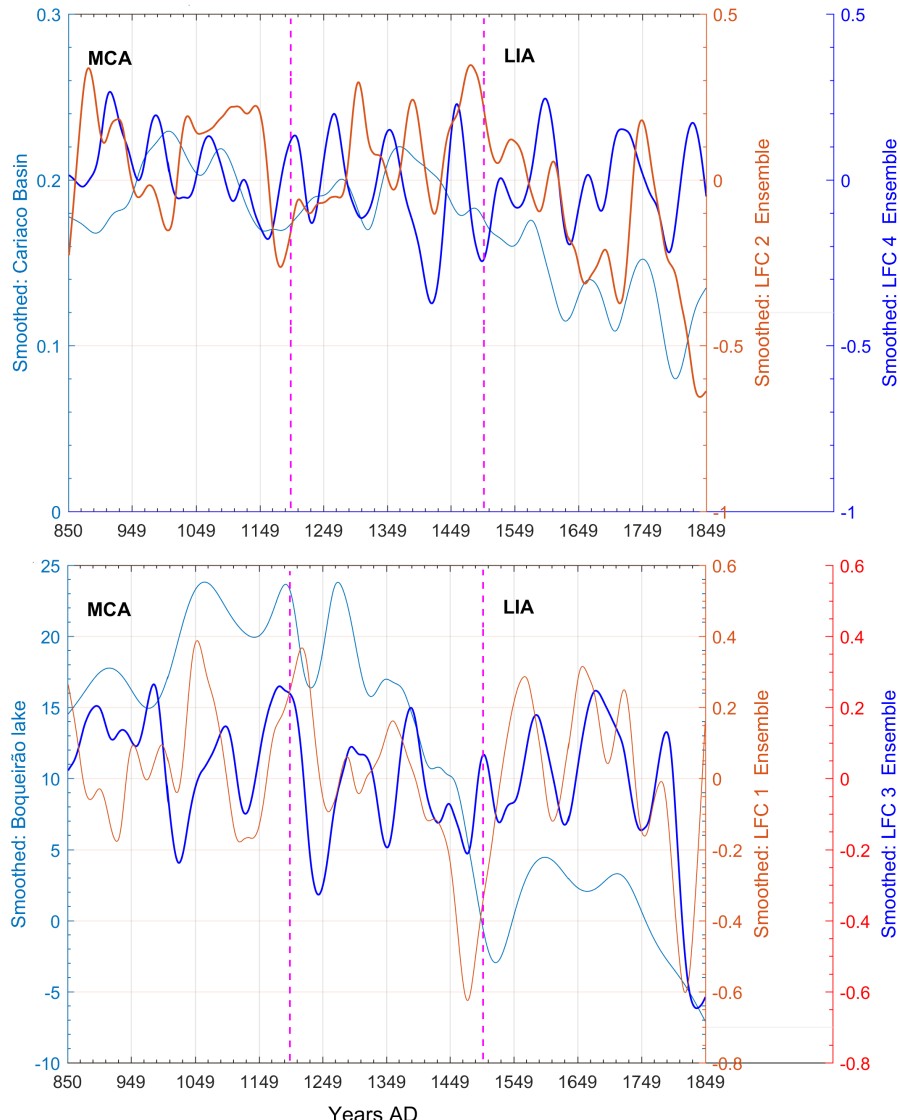

**Figure 4.** HP filter applied to time series (mm/years) with $\lambda = 14{,}400$ of paleoclimate proxies and Low-frequency components (LFCs) precipitation anomalies through the last millennium ensemble in the ensemble (Averaging the models MRI- ESM2-0 and MIROC-ES2L) based on low-frequency component analysis with 30 EOFs retained and a 10-year low-pass cutoff. (a) $\delta$Dwax Boqueirão (line light blue)(Viana et al., 2014), LFC2 (line red) and LFC4 (line blue); (b) Ti Cariaco basin (line light blue)(Haug et al., 2001), LFC1 (line red) and LFC3 (line blue) of the precipitation ensemble.

This variability is related to the fact that the simulated precipitation over ocean (land) is overestimated (underestimated). In this context, the climate modeling community has attempted to reproduce the observed ITCZ but even the latest models still fail to depict this phenomenon well (De Szoeke and Xie, 2008; Song and Zhang, 2016; Oueslati and Bellon, 2015). The double





bias of ITCZ undermines the credibility of the application of climate models to climate variability, forecasting and projection studies. Zhou and Xie (2015) noted that during February-April, when the ITCZ climatological zone moves astonishingly to the Southern Hemisphere, the model overestimates (underestimates) the predicted increase in precipitation in the warmer climate of the South (North) of the equator over the Eastern Pacific.

### 4.3 Wavelet coherence analysis

Local and global wavelet coherence spectra show that the components of LFCs precipitation anomalies through the last millennium are coherent with the AMO index at a variety of timescales (Fig. 5). Solid black lines indicate significant coherence at the 5% level. The analysis of the Figures 5 detected, for LFCs, high signal coherence for periodicities in the range of 2 to 4 years with a correlation coefficient of 0.9. This periodicity occurred intermittently throughout the analysis period. It is important to highlight that, in this periodicity range, the arrows pointing upwards (in the 90° direction) indicate that the second signal (i.e.,
AMO) is out of phase by 90° with respect to the first signal (LFCs).

Over the last millennium, the LFC1 (Fig. 5a) exhibits significant periodicity in the 24 and 64 years multidecadal variability during the periods; 949 – 1049 and 1399 – 1549 respectively, when higher frequency variability in the 2 – 8 years range persists throughout the entire period for components LFC 1 and LFC 3 (Figs. 5a, c). For the LFC4 (Fig. 5d) the 12 years multidecadal variability is presented during the period 1049 – 1060, and the years 1265 and 1460. The 21 and 128 years multidecadal
periodicities, during periods 1249 – 1270, when the 4 years higher frequency variability for the LFC2 persists throughout the entire period, yes, for the LFC4 it is persistent from period 1180 – 1849.

The peaks were inferred from the significance of global coherence as shown in (Fig. 5e, f). For the LFC3 component (Fig. 3c) the multidecadal variability 24, 64 and 160 years, Observing (Fig. 5f), it is possible to see the periodicity of 24 years the vector orientations showed the AMO series was advanced 135° between 1200 – 1349 indicating that rainfall responded by 3/8
of the period.

For the periodicity of 64 years during the periods: 949 – 1249 and 1399 – 1549, the vectors showing alterations in their orientations, that is: the 135° Advanced Rainfall Series of the AMO Series, indicating that the AMO responds with 3/8 of the period. A core is observed in which the AMO series is out of phase by 90° of precipitation. Answering the AMO in 1/4 of the period. Another nucleus where the AMO out of phase 45° of the precipitation. That is, the AMO responds in 1/8 of the
period. For the 160-year periodicity, the AMO series is 90° out of phase with the precipitation. Answering the AMO in 1/4 of the period.

For the components LFC2 and LFC4 (Fig. 5b, d) the significant 80-year multidecadal periodicities during the p eriod 1049 – 1349 the vectors have a phase angle equal to 0°, thus indicating that the precipitations are in phase with the AMO, that is, when there was a significant increase (peak) of precipitation, it also occurred in the AMO variability. For the period 1240 – 1349 the
vector orientations have a phase angle equal to 180°, the precipitation and AMO series are completely in opposite phases.

The variability in the range (2–8 years) seems more amplified during the study period (Fig. 5), suggesting that high frequency events were more influential on the climate during this time. La Niña in the variability of the AMO. The warm (cold) phase of the AMO is associated with wetness (drought) condition predominantly over the region considering AMO SST variation





**Figure 5.** (a) Wavelet coherence of AMO indices during last millennium in the ensemble (Averaging the models MRI-ESM2-0 and MIROC-ES2L) (a) LFC1, (b) LFC2, (c) LFC3, and (d) LFC4 Low-frequency patterns precipitation. Thick black contours in wavelet power spectra enclose areas of 5% significance against a red-noise background. Light shading represents the cone of influence. In the global power spectra, thick black lines represents the global wavelet power estimates and thin dashed black lines are the 95% confidence bounds against red-noise background spectra. Periods of peaks in the global wavelet coherence spectra exceeding 95% confidence are indicated.

(Vásquez et al., 2018), it would be reasonable to conclude that higher (lower) variability in Pacific El Niño 3.4 SST trends,

with more frequent EN (LN) events, smaller (more) neutral years precipitation over NEB within shorter (longer) positive and negative trend periods, could be expected during the cold (warm) AMO phase. The analysis of global coherence revealed a





high frequency intensity of 64 years between the periods 949 and 1249 and between 1399 and 1549. These frequencies have also been found in other paleoenvironmental reconstruction works along the Andes (Baker et al., 2005; Novello et al., 2012; Apaéstegui et al., 2014) and are related to variability modes currently recorded in the Atlantic and Pacific Oceans respectively
(AMO - PDO).

The 24 years attendance presents a lot of energy during the transition period between the MCA and LIA. This signal interacts significantly with the frequency of 8 years providing possible evidence of the mechanisms that governed climatology during this period of time. On the other hand, it is evident that the signal with the greatest energy and persistence in the data series is around 64 years during the MCA.

During MCA a relatively low frequency is observed and centered at approximately 80 and 160 years. In the transition period between LIA and MCA after 1200, the 80 years periodicity is maintained and the low-frequency signal decreases, with a high variability appearing around 8 years. During LIA, periodicities of high statistical significance are found around 8, 24 and 64 years; these cycles are persistent.

## 4.4 The Interdecadal Component of the ITCZ latitudinal location

It was observed SST anomalies in the tropical North and South Atlantic Ocean simultaneously affect the position of the ITCZ and consequently the influence of the distribution of precipitation during last millennium. The first four LFP show a warming pattern of SST anomalies in the tropical Atlantic Ocean. It is observed that the latitudinal positioning of the ITCZ would be in the direction of the meridional inter-hemispheric gradient (Tropical Atlantic Dipole) of SST anomalies in the Atlantic (Nobre and Shukla, 1996; Chang et al., 1997; Enfield and Mayer, 1997; Enfield et al., 1999; Wang, 2002; Giannini et al., 2004; Servain
et al., 2014; Schneider et al., 2014; Fechine, 2015). In response to the warming on the SST in the Atlantic and the variation in the intensity of the trade winds, the ITCZ shows a meridional shift (Back and Bretherton, 2009; Tokinaga and Xie, 2011)

In (Fig. 6) LFP1 – Tropical Atlantic warming with contracted south ITCZ South Tropical Atlantic SST is warmer than the North Atlantic. There is a south migration of the ITCZ and NEB is wetter. The lowest pressure area is located in the precipitation region close to NEB compressed by the highest pressure areas. According to the wavelet, the LFP1 is dominated
by interdecadal frequencies (2 – 8 years), is predominantly wet (fig. 5a and e). This relationship suggests that, although ENSO is the main driver in variability over Tropical South America at interannual time scales, this influence can be significantly modulated by Atlantic Ocean climate variability at longer time scales. Some paleoclimatic evidence suggests that solar-force modulation of ENSO appears to be consistent with correlations between precipitation and solar irradiance that are similar to ENSO-related precipitation anomalies (Graham et al., 2007). White and Liu (2008a, b) noted a warm event like El Niño in
the tropical eastern Pacific SST that coincides with peaks in solar force of the ITCZ is, in part, linked to solar activity with a southernmost position of the ITCZ during centenary scale intervals of low solar activity (Poore et al., 2004). This result is consistent with that observed in the West African lake records, where they observed change in precipitation during periods of decrease irradiance (Russell and Johnson, 2007).

In exchange, they vary separately as a North Atlantic mode LFP2 (Fig. 7) There is no SST difference in the equatorial region,
however precipitation is located in a northward position, suggesting a north migration of the ITCZ. In addition, it is observed





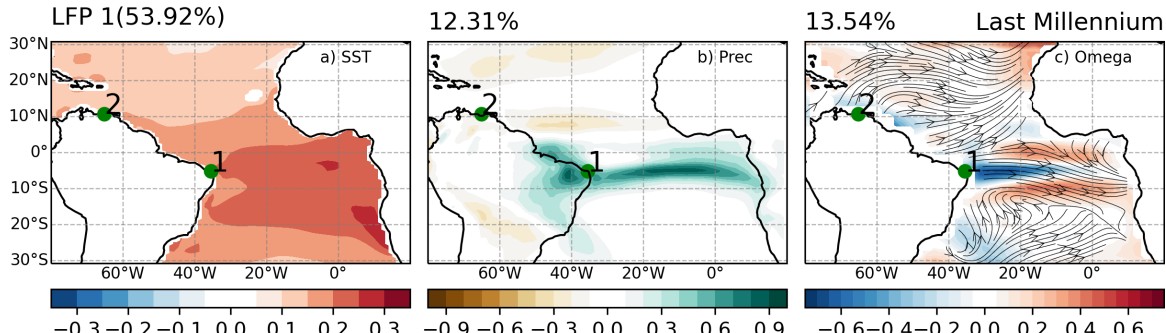

**Figure 6.** First component of Low-frequency patterns (LFP1) sea surface temperature anomalies (°C), precipitation anomalies $\mathrm{mm/month}$ and $500\,\mathrm{hPa}$ omega vertical motion and $850\,\mathrm{hPa}$ through the last millennium in the ensemble (Averaging the models MRI-ESM2-0 and MIROC-ES2L) over the latitudes 30°S to 30°N, based on low-frequency component analysis with 30 EOFs retained and a 10 years low-pass cutoff. The location of the paleoclimate proxies: (1) Boqueirão Lake (Viana et al., 2014); (2) Cariaco Basin (Haug et al., 2001).

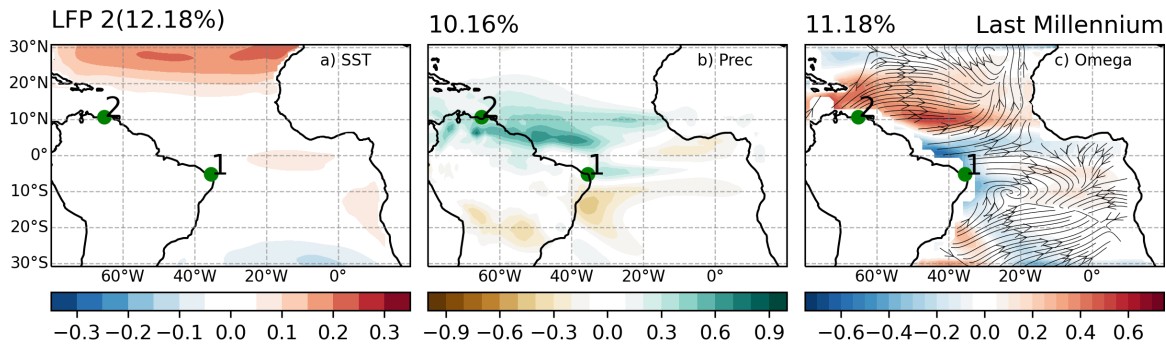

**Figure 7.** Same as figure 6 but for the second component of Low-frequency patterns (LFP2).

that there is no strengthening in southeast winds and weakening of pressure over the South Atlantic close to NEB. The 80 years periodicity is predominant from 1049 to 1350 years, it might be associated with AMO (Figs. 5b and f), also suggested by high SST around 20°–30°N (Fig 6a).

In the second and fourth LFP (Fig. 7 and 8), the mean position of the ITCZ is north of the equator between two regions of
positive and negative SST anomalies. The positive anomalies are found further north of the areas where the greatest differences in precipitation are observed. The most relevant feature was its inclination in the SW-NE direction, extending over practically the entire longitudinal extension of the Equatorial Atlantic, from the north of the NEB to the African continent, in agreement with the results obtained by some authors (Satyamurti et al., 1998; Xavier et al., 2003). The average precipitation range was mostly between 10°S.

In (Fig.8) LFP3 – south ITCZ. There is not a clearly SST difference in the equatorial region, with only a warm tongue between north Africa and Amazon river mouth, however precipitation is located in a southward position, suggesting a south





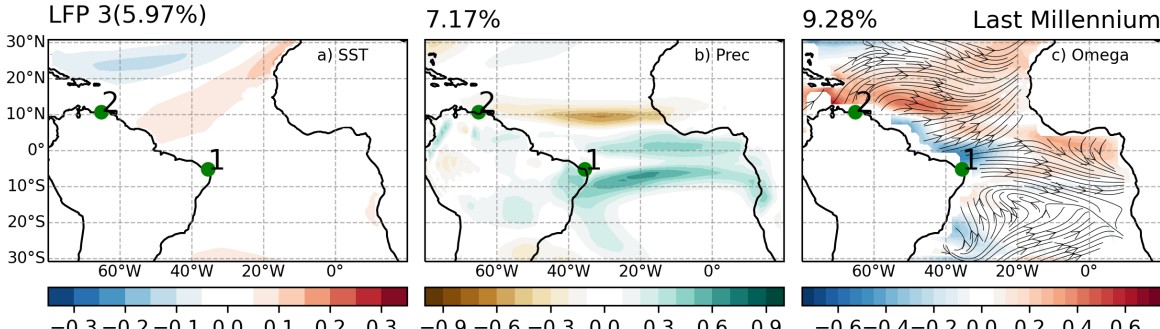

**Figure 8.** Same as figure 6 but for the third component of Low-frequency patterns (LFP3).

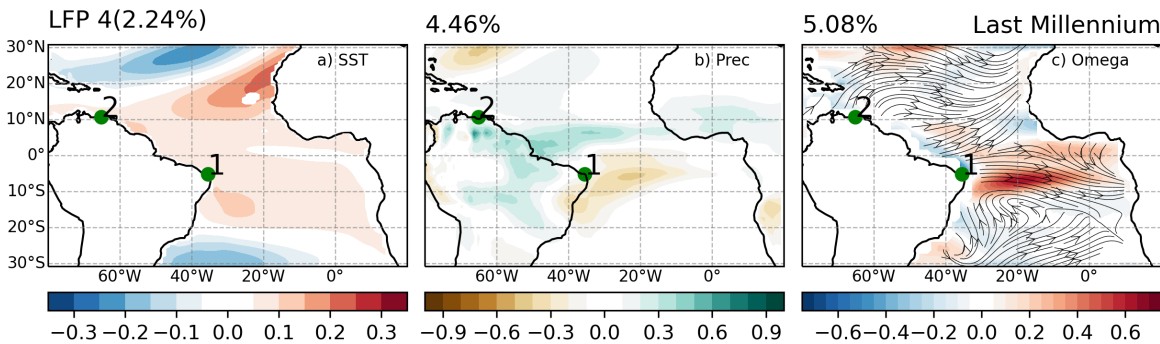

**Figure 9.** Same as figure 6 but for the fourth component of Low-frequency patterns (LFP4).

migration of the ITCZ. The 64 years periodicity is predominant in most parts of the record, except for the last 500 years, that also might be correlated with AMO.

In the omega LFP2 (Fig. 7c) and LFP3 (Fig. 8c) component verified the predominance of the ascending winds between the latitudes 0°–7°N and 4°S–4°N, respectively. However, in the LFP2 component, these upward movements are better defined, in addition to presenting greater extension. The winds at 850 hPa show a west to east flow north of the equator, but in the LFP3 component south of the equator the winds show an opposite flow (from east to west). In the LFP4 (Fig. 9c) component, there is a predominance of downward winds between latitudes 10°S-5°N and in the same region the winds present a flow from east to west. Based on our results, we suggest that this multidecadal variability is mainly due to changes in the AMO phases. Such a mechanism involves the inter-hemispheric SST gradients modulating the position of the ITCZ in the Atlantic.

In (Fig. 9) LFP4 – Tropical Atlantic warming with central contracted ITCZ. The SST suggests a whole Tropical Atlantic warming associated with increased precipitation from Cariaco to NEB limits and from Amazon to central Brazil, while the eastern NEB was under drought conditions. There is a decrease in pressure in the central Tropical region and South Tropical Atlantic, which might be associated with drought conditions in eastern NEB. The 21 years periodicity is predominant from 1050 to 1250 and can be associated to the solar cycle influencing precipitation (Fig. 5d).



LFP1 (Fig. 6) and LFP4 (Fig. 9), show a largely symmetrical mode with respect to the equator; centers with maximum values between 20°N and 25°S east of 5°W, close to the African coast, with their action centers confined to the north and south of the ITCZ, respectively. In exchange, they vary separately as a North Atlantic mode LFP2 and LFP3. Showing an almost uniform warming pattern of the tropical Atlantic throughout the analysis period, this warming would be associated with the

310 ITCZ-associated convection in 5°S (Fig. 5b,c).

It is observed that the latitudinal positioning of the ITCZ would be in the direction of the meridional interhemispheric gradient (Dipole of the Tropical Atlantic) of SST anomalies in the Atlantic (Nobre and Shukla, 1996; Chang et al., 1997; Enfield and Mayer, 1997; Enfield et al., 1999; Wang, 2002; Giannini et al., 2004; Xie and Carton, 2004; Servain et al., 2014; Schneider et al., 2014; Fechine, 2015; Nobre and Shukla, 1996). Showed that, in the TA, warmer SST are associated with less

315 intense trade winds, while cooler SST are associated with more intense trade winds. Variations in the intensity of the trades appear as the main driver of thermal changes in the ocean surface over the TA, resulting in anomalous patterns of SST. The southern component of the wind appears to be responsible for the anomalous southern gradients of SST in the EA, suggesting that anomalies in atmospheric circulation far from the equator force the formation of anomalies in the southern gradient of SST, which force the ITCZ to follow the displacement of warmer waters, affecting the precipitation distribution in the EA and

320 adjacent areas.

It is generally observed that, during the study period, the components of the trends of the LFCs (Fig. 10 and Fig. 11) have a high variability, being accentuated during the LIA. In relation to the pattern in the TA on the decadal scale, the SST has a more pronounced north-south in the LFP2 (Fig. 9d) dipole became neutral due to the North Tropical Atlantic being colder and the South Tropical Atlantic being slightly colder. This configuration disfavors the distribution of moisture and energy to the

325 atmosphere and, therefore, less systems with precipitant potential.

The LFP1 (Fig. 10a) indicates that the LIA was wetter than the MCA (Fig. 10b) in central Brazil, associated with a monsoon strengthening, with warmer South Atlantic than North Atlantic, however the SST does not show differences higher than 0.2 °C between MCA and LIA (Fig. 11a and b).The extreme NEB is wetter during the MCA than the LIA, suggesting an ITCZ strengthening and located in a southernmost position. The pattern north-south ITCZ was actived during the MCA, when Cariaco

330 was dry and NEB was wet. Although, during the LIA the model does not show great precipitation differences between Cariaco and Boqueirão, as the proxies indicate. In the LFP2 (Fig. 10c) the MCA is wetter in Cariaco than in Boqueirão, however with smaller SST temperatures, which is not expected for the MCA in the northern hemisphere (Fig. 11d). The LIA is wetter in Boqueirão with higher SST than in Cariaco (Fig. 9d), as expected for the South migration of the ITCZ (Fig. 10d).

Positive values of sea surface temperatures were observed in the Gulf of Guinea in the extreme rainy years in the NEB.

335 According to Nobre and Shukla (1996), negative SST anomalies in the northeast Atlantic, close to the coast of Africa, and positive anomalies in the central part of the South Atlantic would significantly affect the position and intensity of the ITCZ and, therefore, exert considerable influence on the pattern of rains in the NEB.

On the other hand, when the waters of the TA North are cooler and the waters at the TA South are warmer (Fig. 11), the ITCZ is located in its southernmost position (Fig. 10b, d, e and f) the formation of a meridional gradient of SST anomalies

340 from north to south is observed. This causes the Sea Level Pressure (SLP) to be less than average over the TA south and higher



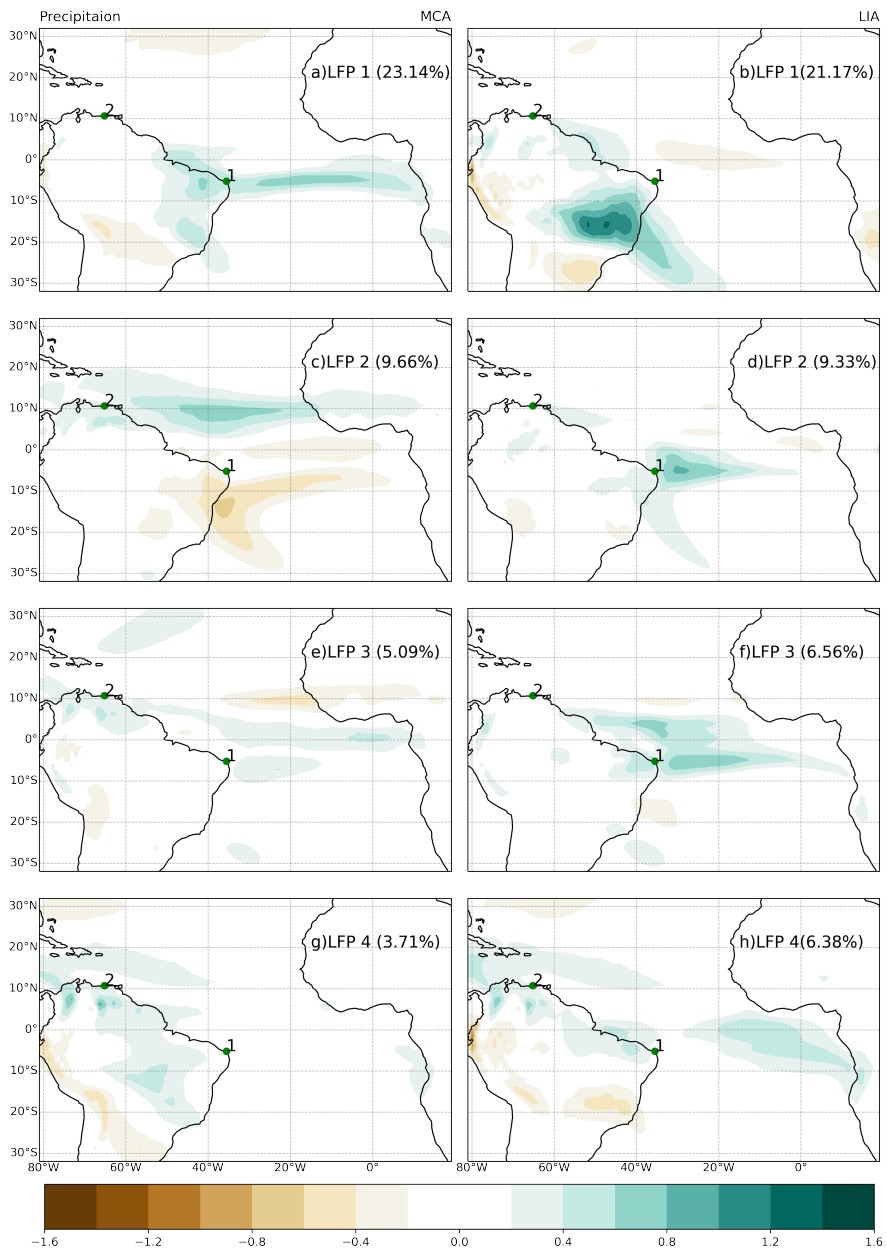

**Figure 10.** Low-frequency patterns (LFPs) precipitation anomalies $\mathrm{mm/month}$ through the MCA (850 – 1200) and LIA (1500 – 1850) in the ensemble (Averaging the models MRI-ESM2-0 and MIROC-ES2L) over the latitudes $30°\mathrm{S}$ to $30°\mathrm{N}$, based on low-frequency component analysis with 30 EOFs retained and a 10-year low-pass cutoff. The location of the paleoclimate proxies: (1) Boqueirão Lake (Viana et al., 2014); (2) Cariaco Basin (Haug et al., 2001).



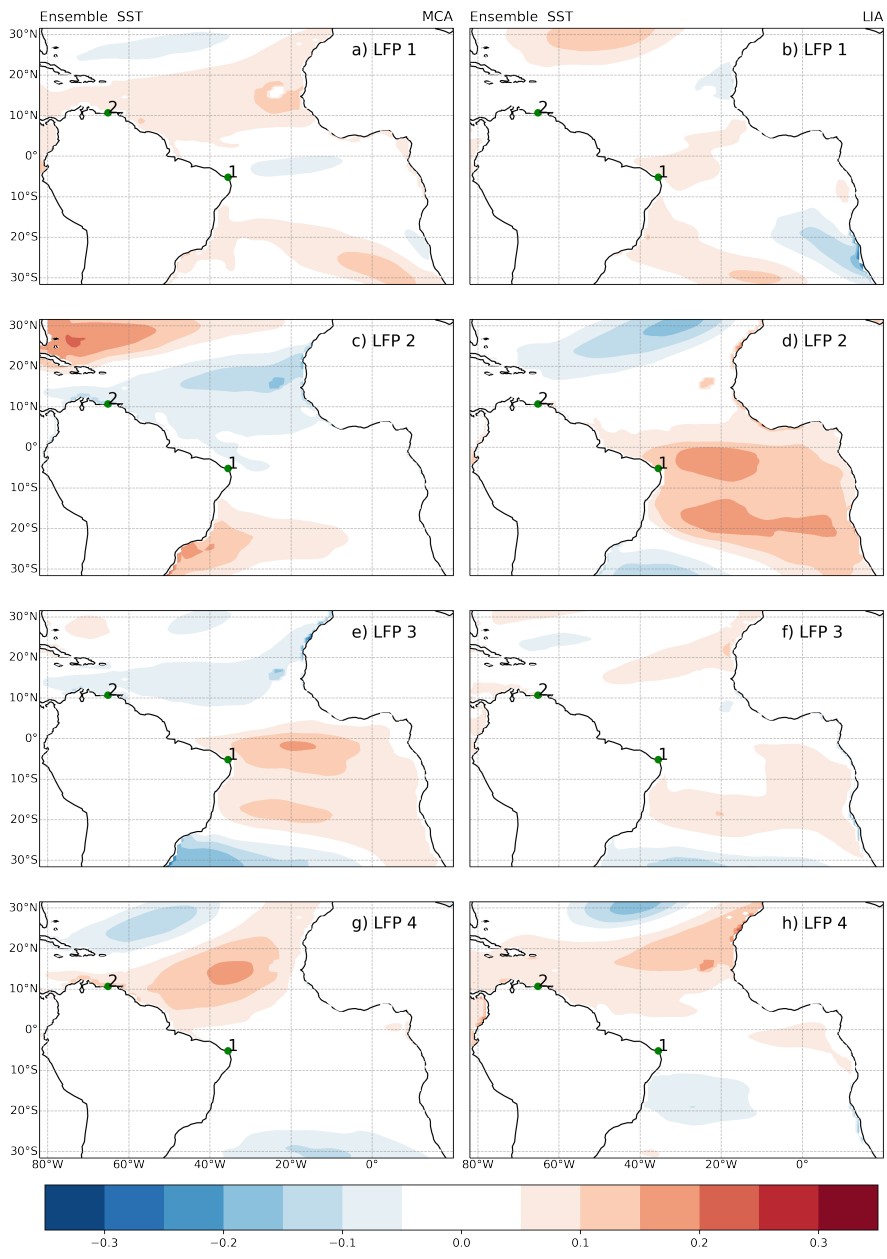

**Figure 11.** Low-frequency patterns (LFPs) sea surface temperature anomalies (°C) through the MCA (850 – 1200) and LIA (1500 – 1850) in the ensemble (Averaging the CMIP6 models MRI-ESM2-0 and MIROC-ES2L) over the latitudes 30°S to 30°N, based on low-frequency component analysis with 30 EOFs retained and a 10 years low-pass cutoff. The location of the paleoclimate proxies: (1) Boqueirão Lake (Viana et al., 2014); (2) Cariaco Basin (Haug et al., 2001).





than average over the AT North, weakening of SE trades and intensification of NE trades and increase of upward movements over the regions to the south, intensifying cloud formation and increasing total rainfall (AMO negative phase) (Hastenrath and Heller, 1977; Moura and Shukla, 1981)

During the MCA precipitation is distributed in the South America Atlantic coast, when Cariaco and Boqueirão are wet (Fig. 10e). The SST is warmer in the whole South Tropical Atlantic (Fig. 11e ). NEB is wetter during the LIA in the LFP3 (Fig. 10f), and precipitation is also over the Amazon coast, however Cariaco is dry. The variation of SST between north and south Atlantic is smaller, but still warmer in the south In the LFP3 (Fig. 11f). The precipitation is increased in NEB during the LIA when compared with MCA (Fig. 10h and g), however in both periods the north Tropical Atlantic has higher SST (Fig. 11g and h). The Cariaco area is wet during the LIA, but the proxy indicate contrary conditions. When the waters of the TA North are warmer

and the waters from the TA South are colder, there are downward movements carrying cold and dry air from the upper levels of the atmosphere over the northern, central and sertão region of the NEB, inhibiting cloud formation and decreasing precipitation (positive phase of the AMO), which can cause droughts. SST anomalies significantly affect the position and intensity of the ITCZ and, can modulating the seasonal distribution of rainfall over the Equatorial Atlantic, from the northern part of the NEB to the central part of Amazonia (Moura and Shukla, 1981; Uvo, 1989; Servain, 1991; Hastenrath and Greischar, 1993; Molion,

1993; Nobre and Shukla, 1996; Marengo, 2004; Marengo et al., 2017; Enfield and Mayer, 1997).

## 5   Conclusions

Our results using the probabilistic method, indicate that the CMIP5 and CMIP6 models generally simulate a more frequent southward migration of the Atlantic ITCZ than the observed one.

The ITCZ structure in the models is strongly influenced by the seasonal cycle of precipitation, the increase in rainfall

observed to be a factor in the growth of the southern bias.The southward shift of the ITCZ in the models was observed in previous studies (Richter and Xie, 2008; Richter et al., 2014; Zermeño-Diaz and Zhang, 2013; Mamalakis et al., 2021), which causes an overestimation of the accumulated precipitation in the NE region of Brazil.

The models CMIP6 and CMIP5 have a tendency to reproduce a double-ITCZ in the Atlantic Ocean (Fig.2 ).This finding is in contrast to recent studies (Samanta et al., 2019; Tian and Dong, 2020; Mamalakis et al., 2021). Considering the reported

biases, the models were able to simulate the seasonal variation of the ITCZ, being able to produce the dominant modes of variability in the position of the ITCZ in the tropical Atlantic.

For the last millennium period, the precipitation of CMIP6 during the MCA and LIA is consistent with paleoclimatic proxies in the Midwest region. In the Northeast region, the MCA period (LIA) was characterized by a decrease (increase) in rainfall. Therefore, the extreme events of the LIA were more intense than in the MCA. Regarding the influences on precipitation

extremes, the indicators also showed strong influences of SAMS, SACZ and the SST under the tropical Atlantic Ocean.

In interannual to decadal scale in both modes, the importance of coupling between the atmosphere and the ocean was observed.The results suggest that the north-south displacement of the ITCZ maintained a relationship with the oceanic region with the highest SST in the tropical basin of South Atlantic. The zonal mode variability is initially associated with the equatorial



region (between 5°S and 5°N) and with the northwestern African coast. The observed frequencies help to dissociate the
mechanisms that affect precipitation in these two areas during MCA and LIA. Periodicities of 64 years during MCA are found
in both records, suggesting that both parts of the continent were affected by the same mechanism that causes dry conditions. For
the LIA period, interactions in different modes (8, 24, 65 years periodicities) bring greater variability in the system, explaining
the increase in ITCZ activity and its regional pattern. Based on the periodicities the 21 years observed is predominant and can be
associated with the solar cycle influence on the pattern of ITCZ contracted and positioned in the central region of the Equator.
This relationship suggests that, although ENSO is the main driver in variability over Tropical South America at interannual
time scales, this influence can be significantly modulated by longer time scales. This multidecadal variability is mainly due
to changes in the AMO phases. Such mechanism involves the inter-hemispheric SST gradients modulating the position of the
ITCZ in the Atlantic.

The results suggest the existence of a low frequency variability, modifying the distribution of precipitation and with conse-
quences in the intensity and frequency of droughts/floods events in the NE, indicating that these events are associated with the
coupling of the oceans to the atmosphere.

*Author contributions.* Vásquez, Gilvan and Humberto designed the study. Vásquez developed and performed the models analysis with help
from Arturo and David. Vásquez wrote the paper with contributions from all co-authors.

*Competing interests.* The authors declare that they have no conflict of interest.

*Acknowledgements.* This study was supported in part by the Grants FAPESP (Fundacão de Amparo à Pesquisa do Estado de São Paulo)
project (Number 2019/08726-7 to Isela Vásquez and 2020/02737-4 to Giselle Utida); Climate Research and Education in the Americas using
Tree-ring and Speleothem Examples (PIRE-CREATE) project (Number 2017-50085-3); CNPq (*Conselho Nacional de Desenvolvimento
Científico e Tecnológico*) project *Programa de monitoramento da desertificação por satélite no Semiárido brasileiro*, under the Grant/Award
Number (403223/2021-0 to H.A.B); RTI-09427-B-C22, Project KUK  AHPAN *Amenaza y Riesgo Sísmico en América Central y Sureste de*
*España* (RTI-09427-B-C22 to J. G. Rejas).



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
