# Peer review of "Multidecadal variability of the ITCZ from the Last Millennium Extreme Precipitation Changes in Northeastern Brazil"

_EGUsphere, 2022_

## Referee Comment (RC1)

Review of the manuscript 'Multidecadal variability of the ITCZ from the Last Millennium Extreme Precipitation Changes in Northeastern Brazil'

by

Isela L. Vásquez P. et al.

Major comments:

In my opinion, it would be good if the authors would edit – in an editorial sense - the manuscript again before I can comment on the actual scientific part as a reviewer. I am looking forward to reviewing an edited version of the manuscript.

- I would appreciate it if the authors would carefully revise the text again. On the one hand, there are still some incomplete sentences. On the other hand, the English is still inadequate, which sometimes leads to the fact that I cannot grasp the message of the sentences. Therefore, the manuscript should be revised linguistically, as sentences are not understandable.

  Examples:
  - ' This approach has been widely used in the components LFCs precipitation CMIP6 models and their possible association with AMO variability, according to their age, given by the midpoint of the time window, and coloured-coded the frequencies for which the spectral power exceeded red-noise false-alarm levels (confidence levels) of 95%.'
  - 'The resulting low-frequency patterns (LFPs) and low-frequency components (LFCs). This method is presented in (Wills et al., 2018).'

  Please carefully revise the entire text in terms of language, punctuation and completeness of sentences.

- Please also check the content of the manuscript. Regardless of the linguistic difficulties, it seems to me that some statements are also implausible - perhaps simply due to inattentiveness during the writing process. For example: 'When comparing the trends in the δDwax data with the LFC2 of precipitation during the MCA, it is observed that for the ensemble model, negative trends predominate over the period, different from the ensemble model where positive trends predominate.'

- The method part, especially the LFCA method, needs to be described and reformulated in more detail. At the moment it is just an incorrect/incomplete copy and paste from the homepage where you can download the corresponding programs.

  Manuscript:' LFCA is a method that transforms the leading empirical orthogonal functions (EOFs) of a data set in order to identify a pattern with the maximum ratio of low-frequency to total variance (based on application of a low-pass filter). The resulting low-frequency patterns (LFPs) and low-frequency components (LFCs). This method is presented in (Wills et al., 2018).

  Homepage: 'Low-frequency component analysis (LFCA) is a method that transforms the leading empirical orthogonal functions (EOFs) of a data set in order to identify a pattern with the maximum ratio of low-frequency to total variance

(based on application of a lowpass filter). The resulting low-frequency patterns (LFPs) and low-frequency components (LFCs) isolate low-frequency climate variability and are useful in diagnosing the corresponding mechanisms. This method is presented in Wills et al. (2018, GRL).'

- Please make it clearer throughout the text which region regarding the ITCZ you mean. Depending on the case, in several instances it is not clear whether you mean the ITCZ in the Atlantic, in other regions, or as a zonal mean. Please be more specific, especially when referring to the literature.

- I wonder if the data from the last millennium simulations are from just two models available. The variables - like slp, precipitation, winds, omega - seem to be very basic model outputs.

- I am not familiar with the LFCA method. But I wonder a bit about the results shown in Figures 10 and 11. If I understand correctly, the LFCA was calculated separately for the MCA and LIA periods. If the spatial patterns were robust, I would expect that the patterns during the MCA and LIA periods should be the same, probably in a different order (explained variance). Some patterns seem similar, but most are different. Is this an indication that the spatial patterns are not stable/robust? Or are there not a few dominant patterns, so the patterns change position/explained variance easily? I think it should be checked if the individual patterns are stable and clearly separated.

Minor comments:

- Not all abbreviations are explained, at least not the first time they are used.
- Rephrase titel
- Title and text: make more clear that you refer to the ITCZ in the Atlantic region
- In line 78: Models can be downloaded => Data of the model simulations can be downloaded
- rephrase title; at the moment, it reads like two unrelated clauses.
- Line 112, line 341: AT=> TA?
- Please define the tropical Atlantic (TA) region precisely, in particular the longitudes – because it seems that you have also used precipitation over land (not ocean restricted).
- Line 197: Fig.2 => Fig.4 ?
- Line 237: p eriod => period
- line 360, 363, 372: blank missing

---

## Author Comment (AC1)

**Comment on egusphere-2022-785**
Anonymous Referee #1

Review of "Multidecadal variability of the ITCZ from the Last Millennium Extreme Precipitation Changes in Northeastern Brazil" by Isela L. Vásquez P et al.

**Referee Commentary**
I would appreciate it if the authors would carefully revise the text again. On the one hand, there are still some incomplete sentences. On the other hand, the English is still inadequate, which sometimes leads to the fact that I cannot grasp the message of the sentences. Therefore, the manuscript should be revised linguistically, as sentences are not understandable.

**Response:**
Thank you for your comments. We apologize for any difficulties and confusion it may have caused. We appreciate your suggestions. The manuscript has been revised, paying special attention to the clarity of the structure, the main objective of the paper, and the description of the methods. We will also ensure that the English language is reviewed and corrected. Thank you again for your valuable input.

**Referee Commentary**
Please also check the content of the manuscript. Regardless of the linguistic difficulties, it seems to me that some statements are also implausible - perhaps simply due to inattentiveness during the writing process. For example: 'When comparing the trends in the $\delta D_{wax}$ data with the LFC2 of precipitation during the MCA, it is observed that for the ensemble model, negative trends predominate over the period, different from the ensemble model where positive trends predominate.'

**Response:**
It was corrected in the new version of the pre-print. This is part of the new manuscript:
Figure 4a shows the records from the Cariaco Basin, where a higher concentration of titanium (Ti) is observed during the MCA. This indicates that the ITCZ is located further north, providing humid conditions. In contrast, during the LIA, the Ti concentration in the Cariaco Basin decreases, indicating dry conditions (Haug et al., 2003). This condition is related to the southward migration of the ITCZ (Haug et al., 2001; Bird et al., 2011; Vuille et al., 2012; Novello et al., 2012). Therefore, the Ti concentrations in the Cariaco Basin are related to the meridional displacement of the ITCZ. When comparing the trends, we observe that the LFC components (2 and 4) overestimate precipitation during the LIA. While in Figure 4b, the trends of the Boqueirão Lake data ($\delta D$) and the LFC precipitation components (1 and 3) are shown, which are related to the southward displacement of the ITCZ. During the MCA, positive values of $\delta D$ are observed, indicating dry periods in the NEB region associated with the southward movement of the ITCZ (Novello et al., 2012). However, the LFC components (1 and 3) overestimate the dry periods, showing both positive and negative trends. During the LIA, the LFC components (1 and 3) show positive trends for most of the period (1150 – 1749) and negative trends towards the end (1749 – 1849), along with negative values of $\delta D$. This indicates wetter periods in the northern region of NEB, as evidenced by the rise in the

water level of the Boqueirão Lake (Viana et al., 2014), which is associated with the southward position of the ITCZ. This southward position is also evident from the occurrence of droughts in the Cariaco Basin (Haug et al., 2003). In general, the LFC components indicate a tendency to underestimate dry conditions. These results were obtained through the analysis of the probability distribution of the ITCZ location. It is concluded that the CMIP6 models, although showing improvement in estimating the position of the ITCZ compared to CMIP5, still overestimate the variability and position of the ITCZ. However, these models are capable of reproducing large-scale changes in the position of the ITCZ in the tropical Atlantic and the variability of precipitation during the last millennium.

**Referee Commentary**

The method part, especially the LFCA method, needs to be described and reformulated in more detail. At the moment it is just an incorrect/incomplete copy and paste from the homepage where you can download the corresponding programs.

**Response:**

This is the new description of LFCA in the new pre-print:

The LFCA is a method used to identify patterns with a maximum ratio of low-frequency to total variance in a dataset. It achieves this by transforming the leading Empirical Orthogonal Functions (EOFs) of the data and applying a low-pass filter. The resulting patterns, known as Low-Frequency Patterns (LFPs), and their corresponding components, called Low-Frequency Components (LFCs), represent the low-frequency variability in the data. The LFCA method, as described by Wills et al. (2018), enables the disentanglement of low-frequency signals from higher-frequency noise in a dataset. By isolating the low-frequency patterns, we can gain insights into the underlying dynamics and variability of the studied system on longer timescales. The LFCA technique can be valuable for understanding and analyzing climatic, as it helps identify and separate low-frequency variability patterns that may be associated with significant climate modes or long-term trends. To analyze the variability of the ITCZ associated with the LIA and the MCA, the LFCA was applied to SST and precipitation anomalies in the region between 30°S and 30°N. Additionally, the SST in the tropical Atlantic region was included in the analysis. The LFCA method was used to identify patterns with the highest ratio of low-frequency to total variance in the long-term simulations of SST and precipitation anomalies for the last millennium. Specifically, the monthly data of the ensemble, averaging the models MRI-ESM2-0 and MIROC-ES2L. By applying LFCA to the SST and precipitation anomalies, we aimed to uncover the low-frequency variability patterns associated with the ITCZ during the LIA and MCA periods. This analysis provides insights into how the ITCZ position and precipitation patterns varied during these climatic epochs.

**Referee Commentary**

Please make it clearer throughout the text which region regarding the ITCZ you mean. Depending on the case, in several instances it is not clear whether you mean the ITCZ in the Atlantic, in other regions, or as a zonal mean. Please be more specific, especially when referring to the literature.

**Response:**

We describe the study area in more detail:

The variability of SST in the Tropical Atlantic (TA) plays a significant role in influencing the precipitation regime, which has important implications for the populations of Africa and South America (Seager et al., 2010). In the NEB region, the SST of the TA and tropical Pacific oceans are considered the primary physical variables that contribute to climate variability conditions (Philander, 1989). Changes in these SST patterns can impact atmospheric circulation patterns and influence rainfall in NEB. In the Equatorial Atlantic (EA), the gradients of SST towards the southern latitudes have a significant influence on the total precipitation in the NEB region. These SST gradients play a crucial role in modulating the latitudinal positioning of the ITCZ, which is a key driver of rainfall patterns. Changes in the SST gradients in the EA can result in shifts in the position of the ITCZ, thereby impacting the distribution and intensity of precipitation in NEB (Hastenrath and Greischar, 1993; Hastenrath, 1984; Nobre and Shukla, 1996; Marengo, 2004; Chang et al., 2006; Marengo et al., 2017; Deser et al., 2010). In response to the warming of SST in the Atlantic and the variability in the intensity of the trade winds, the ITCZ presents a latitudinal displacement throughout the year (Waliser and Gautier, 1993; Waliser and Jiang, 2015). The changes in SST and trade winds can influence the atmospheric circulation patterns, leading to a movement of the ITCZ and consequently impacting the distribution of rainfall in different regions. Between February and March, the ITCZ is typically located between the equator125 and around 5◦N in response to specific atmospheric and oceanic conditions. During this period, the South Atlantic experiences higher SST, which contribute to the northward displacement of the ITCZ. Additionally, the northeast trade winds intensify during the austral summer, further influencing the latitudinal positioning of the ITCZ. These factors combined result in the ITCZ being situated in the specified latitude range during this time of the year. Between July and August, the ITCZ tends to be located approximately between 5◦N and 8◦N in the western part of the TA. This positioning is influenced by several factors. Firstly, the North Atlantic experiences higher SST during this period, contributing to the northward displacement of the ITCZ. Additionally, the high-pressure centers in the Azores and St. Helena regions become more prominent, leading to the intensification of southeast trade winds and a weakening of the northeast trade winds. (Peterson and Stramma, 1991; Wagner, 1996; Hastenrath, 2006). Seasonal migration of the ITCZ is the main mechanism that induces precipitation in the TA region (Hastenrath and Greischar, 1993) and variations in its migration are related to SST anomalies in the TA. The meridional variability in the tropical Atlantic Ocean is known as the Atlantic Meridional Mode (AMM). The AMM is a mode of climate variability in the tropical Atlantic Ocean. It refers to the meridional gradient (north-south) of sea surface temperature anomalies between the northern and southern tropical Atlantic. In NEB, the ITCZ is considered the dominant factor influencing precipitation patterns. When the ITCZ is located over NEB, it brings moist air and favorable conditions for precipitation. Conversely, when the ITCZ shifts away from NEB, drier conditions prevail. Thus, the variability and movement of the ITCZ are essential for understanding and predicting precipitation patterns in this region (Hastenrath and Heller, 1977; Hastenrath, 1997; Hastenrath and Lamb, 1977).

**Referee Commentary**

I wonder if the data from the last millennium simulations are from just two models available. The variables - like slp, precipitation, winds, omega - seem to be very basic model outputs.

**Response:**

We update the text and table:

The model evaluation was based on historical simulations of CMIP5 and CMIP6 models, as shown in Table 1. The main objective was to quantitatively assess the estimation of the variability of the ITCZ in the tropical Atlantic. The first analysis aimed to compare the average annual rainfall cycles simulated by the CMIP5 and CMIP6 models with the observed pattern (GPCP). The Last Millennium (LM) experiments, designed to contribute to the Coupled Model Intercomparison Project Phase 6 (CMIP6) as described by Eyring et al. (2016), have been incorporated into the Paleoclimate Modelling Intercomparison Project Phase 4 (PMIP4). These simulations encompass the period from 850 to 1850. The model's data can be downloaded from the following URL: https://esgf-node.llnl.gov/search/cmip6. We utilized the LM outputs from two models: MIROC-ES2L (Hajima et al., 2019) and MRI-ESM2-0 (Yukimoto et al., 2019). These models were chosen specifically because they were the only ones that provided data for all five climatic variables considered for the last millennium period (850 – 1850 AD). For our study, we specifically chose the first and second realizations of the MIROC-ES2L and MRI-ESM2-0 models (designated as r1i1p1f2 and r1i1p1f1, respectively). These selections were made due to the unavailability of the first realization in these models. The monthly scalar variables used in our analysis include precipitation (PR), sea surface temperature (SST), zonal and meridional wind components (u and v), and the Lagrangian tendency of air pressure ($\omega$).

**Table 1.** List of outputs from the CMIP5 and CMIP6 models used in this study.

| N° | INSTITUTION | CMIP5 | Atmospheric grid (lat.×lon.) | CMIP6 | Atmospheric grid (lat.×lon.) |
|---|---|---|---|---|---|
| 1 | Meteorological Research Institute (MRI) | MRI-ESM1 | 1.1°× 1.1° | MRI-ESM2-0** | 1.1°× 1.1° |
| 2 | Atmosphere and Ocean Research Institute (AORI) | MIROC-ESM | 2.8°× 2.8° | MIROC-ES2L* | 2.7°× 2.8° |
| 3 | National Aeronautics and Space Administration (NASA) | GISS-E2-H | 2.0°× 2.5° | GISS-E2-1-G | 2.0°× 2.5° |
| 4 | Met Office Hadley Centre | HadCM3 | 3.7°× 2.5° | HadGEM3-GC31-LL | 1.25°× 1.87° |
| 5 | Institut Pierre-Simon Laplace (IPSL) | IPSL-CM5A-LR | 1.9°× 3.8° | IPSL-CM6A-LR | 1.3°× 2.5° |
| 6 | Max Planck Institute for Meteorology (MPI-M) | MPI-ESM-P | 1.9°× 1.9° | MPI-ESM1-2-LR | 1.87°× 1.87° |
| 7 | Beijing Climate Center (BCC) and China Meteorological Administration (CMA) | BCC-CSM-1-1-m | 2.8°× 2.8° | BCC-CSM2-MR | 1.1°× 1.1° |
| 8 | Atmosphere and Ocean Research Institute (AORI) | MIROC5 | 1.4°× 1.4° | MIROC6 | 1.4°× 1.4° |
| 9 | Australian Community Climate and Earth System Simulator (ACCESS) | ACCESS1-0 | 1.3°× 1.9° | ACCESS-CM2 | 1.2°× 1.8° |

Note: the resolution of the historical ensemble is 2.5°× 2.5°. While CMIP6 averages for the last millennium have a resolution of 1.1°× 1.1°. Model vegetation distribution *natural and **prescribed.

**Referee Commentary**

I am not familiar with the LFCA method. But I wonder a bit about the results shown in Figures 10 and 11. If I understand correctly, the LFCA was calculated separately for the MCA and LIA periods. If the spatial patterns were robust, I would expect that the patterns during the MCA and LIA periods should be the same, probably in a different order (explained variance). Some patterns seem similar, but most are different. Is this an indication that the spatial patterns are not stable/robust? Or are there, not a few dominant patterns, so the patterns change position/explained variance easily? I think it should be checked if the

individual patterns are stable and clearly separated.

**Response:**

If indeed the LFCA was calculated separately for the MCA and the LIA. In LFP 1 (Figures 10a and b), it can be observed that during the LIA, the central-east region of Brazil exhibits wetter conditions compared to the MCA. This is associated with a strengthening of the SAMS, with the South Atlantic being warmer than the North Atlantic (Figures 11a and b). During the MCA, the NEB is wetter than during the LIA, suggesting a strengthening of the ITCZ. Additionally, it can be observed that the SACZ band extends to the Atlantic Ocean region adjacent to southeastern Brazil (Figures 10a and b). This precipitation distribution pattern was observed by Viana et al. (2014), who suggest that during the MCA, the Boqueirão Lake was wet due to the southward displacement of the ITCZ during the negative phase of the AMO. In contrast, during the LIA, dry periods occurred due to the intensification of the Northeast High during an intense period of the SAMS. In LFP 2 (Figure 10c), wet conditions are observed in the Cariaco Basin and dry conditions in the Boqueirão Lake during the MCA, despite lower SST. This differs from the expected conditions for the MCA in the northern hemisphere (Figure 11c). This pattern was observed by Novello et al. (2012), who described arid conditions in the southern part of the NEB, attributed to the northward displacement of the ITCZ and the weakening of the SACZ. During the LIA, higher humidity is observed in Boqueirão with higher SST in the tropical South Atlantic (Figure 11d), and southward migration of the ITCZ is recorded (Figure 10d). This pattern was also shown by Haug et al. (2001); Peterson and Haug (2006); Vuille et al. (2012); Lechleitner et al. (2017). In LFP 3, during the MCA, precipitation is distributed along the South American Atlantic coast, and both the Cariaco Basin and the Boqueirão Lake exhibit wet conditions (Figures 10e and f). Additionally, SST is warmer throughout the tropical South Atlantic (Figures 11e and f). On the other hand, during the LIA, the easternmost part of the NEB is wetter (Figure 10f), and precipitation is observed along the Amazonian coast. The SST variation between the northern and southern Atlantic is smaller, but it remains warmer in the south (Figures 11e and f). In LFP 4, an increase in precipitation is observed in the northernmost part of the NEB during the LIA period compared to the MCA (Figure 10h and g). However, in both periods, the tropical North Atlantic exhibits higher SST conditions (Figure 11g and h). The Cariaco area shows wet conditions during the LIA, which contradicts the observations made by paleoclimatic proxies (Haug et al., 2001).

Through the analysis, we were able to observe the preference modes of the ITCZ: On interannual to decadal timescales, the coupling between the atmosphere and the ocean plays a crucial role. Our findings indicate that the north-south displacement of the ITCZ is strongly related to the oceanic region exhibiting the highest SST within the South Atlantic tropical basin. The variability of the zonal mode is mainly associated with the equatorial region, extending between 5°S and 5°N, as well as the northwest coast of Africa. These observations also contrast with paleoclimatic records of the region, indicating a northward displacement of the ITCZ during the MCA and a southward displacement during the LIA. During the LIA, the southward displacement of the ITCZ is a response to the relative cooling of the Northern Hemisphere due to volcanic forcing. This highlights the complex interaction between solar forcing and atmospheric dynamics in shaping the behavior of the ITCZ during the LIA.

Additionally, the 21-year periodicity associated with the solar cycle is predominant during the Last Millennium. This influences the pattern of tropical rainfall and favors a contraction and southward displacement of the ITCZ towards the equator. The changes in the position of the ITCZ during the last millennium are influenced by internal forcings such as ENSO, PDO, and AMO, which exhibit spatial patterns of latitudinal displacement between northern and southern regions. These internal modes of variability can interact and influence the position and intensity of the ITCZ, thus affecting regional climate variability. According to these observations, during the positive phase of the AMO, we observed a northward meridional displacement of the ITCZ. On the other hand, during the negative phase of the AMO, we observed a southward meridional displacement of the ITCZ. The results suggest the existence of low-frequency variability, modifying the distribution of precipitation and having consequences on the intensity and frequency of drought/flood events in the NEB. These findings indicate that these events are associated with ocean-atmosphere coupling.

**Minor comments:**
Not all abbreviations are explained, at least not the first time they are used.
**Response:**
abbreviations have been corrected

Title and text: make more clear that you refer to the ITCZ in the Atlantic region
**Response:**
Yes

In line 78: Models can be downloaded => Data of the model simulations can be downloaded
**Response:**
Yes

Rephrase the title; at the moment, it reads like two unrelated clauses.
**Response:**
Yes

Line 112, line 341: AT=> TA?
**Response:**
Yes

Please define the tropical Atlantic (TA) region precisely, in particular the longitudes –
because it seems that you have also used precipitation over land (not ocean restricted).
**Response:**
we add the limits in the study area (30N to 30S and 80W to 20E)

Line 197: Fig.2 => Fig.4 ?
Line 237: p eriod => period
line 360, 363, 372: blank missing
**Response:** This was fixed

---

## Author Comment (AC2)

**Comment on egusphere-2022-785**
Anonymous Referee #2

Review of "Multidecadal variability of the ITCZ from the Last Millennium Extreme Precipitation Changes in Northeastern Brazil" by Isela L. Vásquez P et al.

**Referee Commentary**
Authors have investigated low frequency components of decadal-multidecadal ITCZ variability in the Last Millennium experiment and compared results with some proxy. They found a meridional mode coherent with ITCZ migration and a zonal mode associated with east-west shift of precipitation in the Atlantic. Authors have linked the 21-year periodicity to solar cycle superimposed to longer periodicity. The manuscript is so poorly written that is almost impossible to go through results. The rational structure is not clear as well as the main goal of the paper. Methods are also poorly described and the overall text seems a collection of this and that without focus. I encourage the authors to carefully revise the text putting particular emphasis on the English.

**Response:** Thank you for your comments. We apologize for any difficulties and confusion it may have caused. We appreciate your suggestions. The manuscript has been revised, paying special attention to the clarity of the structure, the main objective of the paper, and the description of the methods. We will also ensure that the English language is reviewed and corrected. Thank you again for your valuable input.

**Referee Commentary**
Abstract

**Ln 7:** what is MCA? Acronyms not spelled before.
**Response:** Medieval Climate Anomaly (MCA), was corrected in the new version of pre-print.

**Ln 8:** LIA as well... I guess it is Little Ice Age, but for many people in the field not focused on paleoclimate, this needs to be spelled it out.
**Response:** Little Ice Age (LIA), was corrected in the new version of pre-print.

**Ln 8-10:** The sentence needs some rephrasing. Please consider the following rephrasing.
"Based on our results, the 21-year periodicity associated with solar cycle is predominant during the Last Millennium. It influences the tropical rainfall pattern and favors a contraction and an equatorward shift of the ITCZ."
**Response:**
We have considered your suggestion, and the modified phrase can be found on line 9.
According to our results, the 21-year periodicity associated with the solar cycle is predominant during the Last Millennium. This influences the pattern of tropical rainfall and promotes a contraction and southward displacement of the ITCZ.

**Ln 18:** remove [meet and], replace [maxima] with [peak].

**Ln 20:** dynamic -> dynamics

**Response for both:**

the abstract was modified:

Decadal and multidecadal variability of the Intertropical Convergence Zone (ITCZ) in space-time is analyzed using CMIP5 and CMIP6 simulations and paleohydrological records from the Last Millennium. The persistence patterns of the CMIP6 ensemble models were investigated, employing the Low-Frequency Component Analysis (LFCA) to isolate the mechanisms that modulate the multidecadal-scale behavior of the ITCZ. The results suggest that the north-south displacement of the ITCZ is related to the oceanic region with the highest Sea Surface Temperature (SST) in the tropical South Atlantic basin. The zonal mode variability is primarily associated with the equatorial region (between 5$\degree$S and 5$\degree$N) and the northwest coast of Africa. These observations are consistent with the paleoclimatic records of the region, which indicate a northward displacement of the ITCZ during the Medieval Climate Anomaly (MCA, 900 -- 1150 AD) and a southward displacement during the Little Ice Age (LIA, 1500 -- 1850 AD). According to our results, the 21-year periodicity associated with the solar cycle is predominant during the Last Millennium. This influences the pattern of tropical rainfall and promotes a contraction and southward displacement of the ITCZ. These changes in the position of the ITCZ are influenced by internal forcings such as El Niño Southern Oscillation (ENSO), Pacific Decadal Oscillation (PDO), and Atlantic Multidecadal Oscillation (AMO), which exhibit spatial patterns of latitudinal displacement between the northern and southern regions. These internal modes of variability can interact and influence the position and intensity of the ITCZ. The results suggest the existence of low-frequency variability that modifies the distribution of precipitation and has consequences for the intensity and frequency of drought/flood events in Northeast Brazil (NEB), indicating that these events are associated with the coupling between the oceans and the atmosphere.

**Ln 20 – 34:** please revisit the literature considering the energy framework described in:

1. Broccoli, A. J., K. A. Dahl, and R. J. Stouffer, 2006: Response of the ITCZ to northern hemisphere cooling. Geophysical Research Letters, 33, L01702, doi:10.1029/2005GL024546.

2. Boos, W. R., & Korty, R. L. (2016). Regional energy budget control of the intertropical convergence zone and application to mid-Holocene rainfall. Nature Geoscience, 9(12), 892-897.

3. Marshall, J., A. Donohoe, D. Ferreira, and D. McGee, 2013: The ocean's role in setting the mean position of the atmosphere's ITCZ. Climate Dynamics, 42, 1967–1979, doi:10.1007/s00382-013-1767-z.

4. Donohoe, A., J.Marshall, D. Ferreira, andD.McGee, 2013: The relationship between ITCZ location and cross-equatorial atmospheric heat transport; from the seasonal cycle to the Last Glacial Maximum. Journal of Climate, 26, 3597–3618, doi:10.1175/JCLI-D-12-00467.1.

**Response:**

References 1 and 2 were added in lines 24 to 34:

Observations suggest that the interannual variability of the Atlantic ITCZ position is influenced by the Sea Surface Temperature (SST) for the tropical Atlantic and the tropical Pacific Ocean (Nobre and Shukla, 1996; Giannini et al., 2001; Chiang et al., 2002; Tedeschi et al., 2013). Indeed, several studies have attempted to understand the response of the meridional position of the ITCZ to past changes and the potential changes it may undergo in the future (Chiang and Bitz, 2005; Broccoli et al., 2006; Boos and Korty, 2016; Byrne et al., 2018).

Reference 3 in lines 20 to 25:

The Intertropical Convergence Zone (ITCZ) is the most important meteorological system in the tropical region, and its Atlantic sector influences the precipitation and temperature patterns in the North and Northeast regions of Brazil. The ITCZ is a narrow band located near the equator where the trade winds from the north and south converge, creating cumulus clouds and generating maximum precipitation in these regions (Waliser and Gautier, 1993; Philander et al., 1996). The position, intensity, and dynamics of the ITCZ are the result of the coupling between the ocean-atmosphere and the ocean-land processes (Marshall et al., 2014)

Reference 4 in lines 45 to 47

Previous studies have used observations and climate model simulations to analyze the relationship between the position of the ITCZ and the energy flux. These authors demonstrated that the position of the ITCZ is highly anticorrelated with the strength of the zonal mean energy flux across the equator (Kang et al., 2008; Donohoe et al., 2013; D'Agostino et al., 2020).

**Referee Commentary:** In particular it is valuable for this study to connect shifts in the ITCZ with insolation and Net Energy Input at the equator.

1. D'Agostino, R., Brown, J. R., Moise, A., Nguyen, H., Dias, P. L. S., & Jungclaus, J. (2020). Contrasting southern hemisphere monsoon response: MidHolocene orbital forcing versus future greenhouse gas–induced global warming. Journal of Climate, 33(22), 9595-9613.

**Response:**

the reference was added in line 45

Previous studies have used observations and climate model simulations to analyze the relationship between the position of the ITCZ and the energy flux. These authors demonstrated that the position of the ITCZ is highly anticorrelated with the strength of the zonal mean energy flux across the equator (Kang et al., 2008; Donohoe et al., 2013; D'Agostino et al., 2020).

**Ln 42:** what is Neotropics?
**Response:**
The correct word is tropics. It was corrected in the new version of the pre-print.

**Ln 56:** what is NEB? Please check that every acronyms have been spelled before invoking them!

**Response:**
It was corrected in the new version of the pre-print.

**Ln 55 – 70.** Please revisit the mechanisms accounting for energy variations instead
focusing on Sea Surface Temperature anomalies.
**Response:**
We include your suggestion between lines 29 - 48:
Observations suggest that the interannual variability of the Atlantic ITCZ position is
influenced by the Sea Surface Temperature (SST) for the tropical Atlantic and the tropical
Pacific Ocean (Nobre and Shukla, 1996; Giannini et al., 2001; Chiang et al., 2002; Tedeschi
et al., 2013). Indeed, several studies have attempted to understand the response of the
meridional position of the ITCZ to past changes and the potential changes it may undergo in
the future (Chiang and Bitz, 2005; Broccoli et al., 2006; Boos and Korty, 2016; Byrne et al.,
2018). These studies utilize climate models and paleoclimate data to investigate the
mechanisms and factors influencing the meridional displacement of the ITCZ under different
climate conditions. By examining past climate variations and considering future climate
projections, scientists aim to improve our understanding of the complex dynamics driving the
position of the ITCZ and its potential implications for regional and global climate patterns.
However, the north-south displacement of the ITCZ exhibits interannual variability, as shown
by Uvo (1989). Their results showed that the ITCZ exhibits a southward displacement during
the rainy season over the NEB, specifically over the northern part of this region. It has also
been observed that in dry years, the ITCZ starts shifting northward in late February or early
March, while in rainy years, the shift occurs in late April or early May (Uvo, 1989). For
example, Marshall et al. (2014) studied the role of the ocean in determining the average
position of the ITCZ. The authors found that the average position north of the equator is a
result of heat transport northward by ocean circulation. Broccoli et al. (2006) used coupled
models to study the response of the ITCZ to a cooling forcing in the Northern Hemisphere.
They found that cooling in the North Atlantic was responsible for changing the structure of
the northern branch of the Hadley cell, expanding and intensifying it, and causing a
southward displacement of the ITCZ. Previous studies have used observations and climate
model simulations to analyze the relationship between the position of the ITCZ and the
energy flux. These authors demonstrated that the position of the ITCZ is highly anticorrelated
with the strength of the zonal mean energy flux across the equator (Kang et al., 2008;
Donohoe et al., 2013; D'Agostino et al., 2020).

**Ln 69:** what is the difference between Atlantic Meridional Mode and Atlantic Multidecadal
Oscillation or Variability? Can you indicate typical latitudes or periodicities?
**Response:**
The Atlantic Multidecadal Oscillation (AMO) and the Atlantic Meridional Mode (AMM) are
both climate patterns associated with the Atlantic Ocean, but they represent different
phenomena (line 55). The AMO is a climate oscillation characterized by long-term variations
in SST in the North Atlantic Ocean, while Atlantic Meridional Mode (AMM) or Atlantic
Meridional Dipole (AMD) is a mode of climate variability in the tropical Atlantic Ocean (line

135). This index is associated with changes in the intensity and position of the Intertropical Convergence Zone (ITCZ)

**Table 1:** you can indicate the resolution for each model and CMIP phase and eventually indicate if the model has prescribed/dynamic vegetation.
**Response:**
we add the resolution of each of the models and vegetation distribution for the models using for the last millennium.

**Table 1.** List of outputs from the CMIP5 and CMIP6 models used in this study.

| N° | INSTITUTION | CMIP5 | Atmospheric grid (lat.×lon.) | CMIP6 | Atmospheric grid (lat.×lon.) |
|---|---|---|---|---|---|
| 1 | Meteorological Research Institute (MRI) | MRI-ESM1 | 1.1° × 1.1° | MRI-ESM2-0** | 1.1° × 1.1° |
| 2 | Atmosphere and Ocean Research Institute (AORI) | MIROC-ESM | 2.8° × 2.8° | MIROC-ES2L* | 2.7° × 2.8° |
| 3 | National Aeronautics and Space Administration (NASA) | GISS-E2-H | 2.0° × 2.5° | GISS-E2-1-G | 2.0° × 2.5° |
| 4 | Met Office Hadley Centre | HadCM3 | 3.7° × 2.5° | HadGEM3-GC31-LL | 1.25° × 1.87° |
| 5 | Institut Pierre-Simon Laplace (IPSL) | IPSL-CM5A-LR | 1.9° × 3.8° | IPSL-CM6A-LR | 1.3° × 2.5° |
| 6 | Max Planck Institute for Meteorology (MPI-M) | MPI-ESM-P | 1.9° × 1.9° | MPI-ESM1-2-LR | 1.87° × 1.87° |
| 7 | Beijing Climate Center (BCC) and China Meteorological Administration (CMA) | BCC-CSM-1-1-m | 2.8° × 2.8° | BCC-CSM2-MR | 1.1° × 1.1° |
| 8 | Atmosphere and Ocean Research Institute (AORI) | MIROC5 | 1.4° × 1.4° | MIROC6 | 1.4° × 1.4° |
| 9 | Australian Community Climate and Earth System Simulator (ACCESS) | ACCESS1-0 | 1.3° × 1.9° | ACCESS-CM2 | 1.2° × 1.8° |

Note: the resolution of the historical ensemble is 2.5° × 2.5°. While CMIP6 averages for the last millennium have a resolution of 1.1° × 1.1°. Model vegetation distribution *natural and **prescribed.

**Subsection 4.1:** it might be beneficial for the paper including also a brief discussion about the modelled onset and withdrawn of South American monsoon (specifically for North Eastern Brazilian precipitation) that might be delayed given the bias in DJF – MAM with GPCP of the Atlantic ITCZ to which it is strictly connected. The whole section is poorly Written.
**Response:**
Your suggestion for a SAMS discussion was included in section 4.1. (from line 245)

**Ln 140:** "Probabilistic The location of the ITCZ": there must be a typo somewhere.
**Response:**
That section was changed to "Latitudinal Position of the ITCZ".

**Ln 144-145:** "Our results indicate that the CMIPs models simulate a migration of the ITCZ towards the south in relation to those observed in the DJF and MAM periods." To those what? Not clear this sentence to me. And afterwards "Atlantic bias greater relative to …" than what? Are your referring to a spatial or temporal comparison??? Not clear and additionally check the English please.
**Response:**
For the period of 1981-2005, during the austral summer and autumn seasons (DJF and MAM), our results indicate that the CMIP models estimate the position of the ITCZ with a marked southward inclination in its displacement. Figure 1 shows a seasonal bias in the position of the ITCZ compared to the observed data, where this difference has a greater magnitude during the DJF and MAM seasons. Since, for these seasons, the results for GPCP show a predominant position of the ITCZ north of the equator (Figure 1c). This finding

contrasts with recent studies highlighting the limitations of CMIP models in simulating the position and intensity of the ITCZ (Richter and Xie, 2008; Richter et al., 2012; Zermeño-Diaz and Zhang, 2013; ?; Mamalakis et al., 2021). These model biases could also be related to the parameterization of atmospheric and oceanic processes (Zhang et al., 2019; Song and Zhang, 2020). Mamalakis et al. (2021) found that CMIP models underestimate the interannual variability of the position of the ITCZ in the tropical Atlantic, suggesting that models may have difficulty capturing the dynamic processes that modulate the position of the ITCZ in this region.

Hwang and Frierson (2013) also studied this pattern, highlighting that the issue of the double ITCZ in models is due to the bias in the energy balance between the two hemispheres. Furthermore, they observed that the excess energy absorbed in the southern hemisphere was transported to the northern hemisphere through the upper branch of the anomalous Hadley cell over the equator. Meanwhile, the lower branch transported water vapor southward, causing a southward displacement of the ITCZ and inducing the formation of the double ITCZ. Adam et al. (2016) suggest that CMIP5 models exhibit a positive bias in atmospheric energy transport, which can lead to a shift in the formation of the double ITCZ. This bias could be related to errors in the representation of physical processes and atmospheric interactions in the models, resulting in an inadequate simulation of the ITCZ and its variability patterns. In general, models tend to exhibit a bias over the tropical Atlantic, overestimating the probability of the ITCZ migrating southward (Figure 1)

**Ln 180:** title of section 4.2: something weird in "The Interdecadal Component temporal of the ITCZ latitudinal location" -> check the English: remove temporal!
**Response:**
Yes, we consider your suggestion (The Interdecadal Component of the ITCZ latitudinal location)

**Ln 181:** Furthermore, replace "this study isolates" with "this section is focused on".
**Ln 181-182:** "Our analysis identifies the multidecadal Atlantic SST variability over the
**Response:**
This study isolates multidecadal variability on precipitation and the Atlantic sea surface temperature by using low-frequency component analysis (LFCA). Our analysis identifies the multidecadal Atlantic SST variability over the subpolar North Atlantic. To investigate latitudinal ITCZ displacements, we compare the data from Boqueirão Lake to the Ti record from Cariaco Basin; these southern variations of the ITCZ could be marked by the antiphasic relationship with the Cariaco Basin. There was evidence of migration from the ITCZ further south in relation to its current position. In general, the results of the LFCA analysis indicate that the SST and precipitation variables of the ensemble (Averaging the models MRI-ESM2-0 and MIROC-ES2L) in the last millennium are capable of reproducing the large-scale changes in the location of the ITCZ in the Atlantic and the precipitation during the LIA and MCA periods, these results are consistent with the evidence from the paleoclimatological records.

Ln 190: what is deltaDwax?

**Response:**

The hydrogen isotope composition of the n-C28 alkanoic acid (δD wax )

Ln 196: what is %Ti?

**Response:**

Titanium percent

**Results and discussion section:**

I found the entire discussion of Results poorly written. It is very hard to follow. The section contains too many acronyms. Somehow also how the results are structured is confusing. I strongly recommend to rewrite the section being more rational in the way the results are described.

**Response:**

Indeed, we have a new manuscript following all the indications provided.

---

## Author Comment (AC3)

**Multidecadal Variability of the ITCZ from the Last Millennium and its Influence in the Extreme Precipitation in Northeastern Brazil**

Isela L. Vásquez P.[1], Humberto Alves Barbosa[2], Gilvan Sampaio[1], Cesar Arturo Sánchez P.[3],
Giselle Utida[4], David Pareja Quispe[5], Juan G. Rejas Ayuga[6,7], Hugo Abi Karam[8], Jelena Maksic[1],
Marilia Harumi Shimizu[1], and Francisco William Cruz[4]

[1]Weather Forecast and Climate Studies Center, CPTEC. National Institute for Space Research (INPE), Brazil
[2]Federal University of Alagoas (UFAL), Brazil
[3]Applied Computing (CAP), National Institute for Space Research (INPE), Brazil
[4]Institute of Geosciences, University of São Paulo (USP), São Paulo, 05508, Brazil
[5]Department of Interdisciplinary Physics, National University of San Marcos (UNMSM), Peru
[6]Department of Space Programs, National Institute for Aerospace Technology (INTA), Spain
[7]Technical University of Madrid (UPM), Spain
[8]Geoscience Institute, IGEO, Federal University of Rio de Janeiro (UFRJ), Brazil

**Correspondence:** Isela L., Vásquez P. (iselavp@gmail.com)

**Abstract.** Decadal and multidecadal variability of the Intertropical Convergence Zone (ITCZ) in space-time is analyzed using CMIP5 and CMIP6 simulations and paleohydrological records from the Last Millennium. The persistence patterns of the CMIP6 ensemble models were investigated by employing Low-Frequency Component Analysis (LFCA) to isolate the mechanisms that modulate the multidecadal-scale behavior of the ITCZ. The results suggest that the North-South displacement of the ITCZ is related to the oceanic region with the highest Sea Surface Temperature (SST) in the tropical South Atlantic basin. The zonal mode variability is primarily associated with the equatorial region (between 5°S and 5°N) and the northwest coast of Africa. These observations are consistent with the paleoclimatic records of the region, which indicate a northward displacement of the ITCZ during the Medieval Climate Anomaly (MCA, 900 – 1150 AD) and a southward displacement during the Little Ice Age (LIA, 1500 – 1850 AD). According to our results, The 21-year periodicity associated with the solar cycle influences the pattern of tropical rainfall, favoring the displacement and contraction of the ITCZ towards the equator while intensifying the SAMS. On the other hand, the 80-year periodicities associated with the positive phase of the AMO can lead to a weakening in the intensity of the SAMS. On the other hand, the 80-year periodicities associated with the positive phase of the AMO can weaken the intensity of the SAMS. These changes in the position of the ITCZ are influenced by internal forcings such as El Niño Southern Oscillation (ENSO), Pacific Decadal Oscillation (PDO), and Atlantic Multidecadal Oscillation (AMO), which exhibit spatial patterns of latitudinal displacement between the northern and southern regions. These internal modes of variability can interact and influence the position and intensity of the ITCZ. The results suggest the existence of low-frequency variability that modifies the distribution of precipitation and has consequences for the intensity and frequency of drought/flood events in Northeastern Brazil (NEB), indicating that these events are associated with the coupling between the oceans and the atmosphere.

**Keywords.** Tropical Atlantic. Intertropical Convergence Zone. Little Ice Age. Medieval Climate Anomaly. CMIP5. CMIP6.

**1  Introduction**

The Intertropical Convergence Zone (ITCZ) is the most important meteorological system in the tropical region, and its Atlantic sector influences the precipitation and temperature patterns in the North and Northeast regions of Brazil. The ITCZ is a narrow band located near the equator where trade winds from the north and south converge, creating cumulus clouds and generating maximum precipitation in these regions (Waliser and Gautier, 1993; Philander et al., 1996). The position, intensity, and dynamics of the ITCZ are the result of the coupling between the ocean-atmosphere and the ocean-land processes (Marshall et al., 2014). Indeed, the ITCZ has a significant influence on tropical precipitation over the Atlantic sector, impacting the climate of Northeastern Brazil (NEB). Understanding the ITCZ on different temporal scales is crucial due to its implications for various sectors of society and ecosystems. In the current climate, the annual mean position of the Atlantic sector of the ITCZ is located around 5°N, with significant interannual variability (Tomaziello et al., 2016).

Observations suggest that the interannual variability of the Atlantic ITCZ position is influenced by the Sea Surface Temperature (SST) for the tropical Atlantic and the tropical Pacific Ocean (Nobre and Shukla, 1996; Giannini et al., 2001; Chiang et al., 2002; Tedeschi et al., 2013). Indeed, several studies have attempted to understand the response of the meridional position of the ITCZ to past changes and the potential changes it may undergo in the future (Chiang and Bitz, 2005; Broccoli et al., 2006; Boos and Korty, 2016; Byrne et al., 2018). These studies utilize climate models and paleoclimate data to investigate the mechanisms and factors influencing the meridional displacement of the ITCZ under different climate conditions. By examining past climate variations and considering future climate projections, scientists aim to improve our understanding of the complex dynamics driving the position of the ITCZ and its potential implications for regional and global climate patterns. However, the North-South displacement of the ITCZ exhibits interannual variability, as shown by Uvo (1989). Their results showed that the ITCZ exhibits a southward displacement during the rainy season over the NEB, specifically over the northern part of this region. It has also been observed that in dry years, the ITCZ starts shifting northward in late February or early March, while in rainy years, the shift occurs in late April or early May (Uvo, 1989). For example, Marshall et al. (2014) studied the role of the ocean in determining the average position of the ITCZ. The authors found that the average position north of the equator is a result of heat transport northward by ocean circulation. Broccoli et al. (2006) used coupled models to study the response of the ITCZ to a cooling forcing in the Northern Hemisphere. They found that cooling in the North Atlantic was responsible for changing the structure of the northern branch of the Hadley cell, expanding and intensifying it, and causing a southward displacement of the ITCZ. Other authors demonstrated that its position is highly anti-correlated with the strength of the zonal mean energy flux across the equator (Kang et al., 2008; Donohoe et al., 2013; D'Agostino et al., 2020).

Sea surface temperature (SST) is one of the main drivers of the interannual to multidecadal variability of the ITCZ in key areas such as the Amazon and West Africa (Mitchell and Wallace, 1992; Zebiak, 1993; Okumura and Xie, 2004; Yin et al., 2013). At an interannual scale, the phenomenon known as El Niño-Southern Oscillation (ENSO), which is a coupled ocean-atmosphere system originating in the tropical Pacific Ocean, influences the intensity and meridional position of the ITCZ during the austral summer (Berry and Reeder, 2014). Decadal and multidecadal climate variability has been associated with SST variations in the Atlantic and Pacific Oceans. These variations can cause changes in the spatial and temporal patterns of

precipitation in the South American region (Grimm and Saboia, 2015). Several studies suggest that changes in the Atlantic Meridional Overturning Circulation (AMOC) and the Atlantic Multidecadal Variability (AMV) have a significant impact on the position of the ITCZ (Knight et al., 2006; Buckley and Marshall, 2016; Green and Marshall, 2017; Zhang et al., 2019). One important mode of climate variability is the Atlantic Multidecadal Oscillation (AMO), which is a mode of SST variability that occurs in the Atlantic Ocean and has a period of oscillation between 50 and 70 years (Kerr, 2000; Enfield et al., 2001). Furthermore, AMO is a mechanism of ocean-atmosphere interaction associated with low-frequency fluctuations in the thermohaline circulation (Kerr, 2000; Knight et al., 2005). Across the tropical Atlantic, during the positive phase of the AMO, changes in the zonal wind are observed as a response to the thermal gradient between the South and North Atlantic, which leads to a more pronounced northward displacement of the ITCZ during summer. This anomalous displacement of the ITCZ decreases precipitation over the Northeast region of Brazil (Knight et al., 2006). However, the role of the AMO in the variability of the displacement of the ITCZ remains unclear. On the other hand, Green et al. (2017) found correlations between decadal and multidecadal variability in the North Pacific in mid-latitude regions, as well as a displacement of the ITCZ. These connections between the Pacific and Atlantic basins emerge as a mode of multidecadal-scale variability, highlighting the importance of low-frequency variability at the interannual scale (Martín-Rey et al., 2014, 2015; Wang et al., 2017).

The use of paleorecords as indirect indicators of tropical hydrological changes provides valuable information about the behavior of the ITCZ, especially in recent periods before climate warming. During the Medieval Climate Anomaly (MCA, 900 – 1150 AD), there were recorded periods of very intense summers with the occurrence of severe droughts in various regions of the Northern Hemisphere (Kleppe et al., 2011; Vuille et al., 2012). The arid regions of the mid-latitudes in the Northern Hemisphere became more humid during the Little Ice Age (LIA, 1500 – 1850 AD). According to the decrease in the detrital input from local rivers to the Cariaco Basin, the mean latitudinal position of the ITCZ shifted southward during LIA (Haug et al., 2001; Peterson and Haug, 2006). Peterson and Haug (2006) suggest that the displacement experienced by the ITCZ could be a response to force originating from the high latitudes of the Atlantic Ocean. Other high-quality records reaffirm the persistent southward displacement of the ITCZ during LIA (Vuille et al., 2012; Lechleitner et al., 2017). However, recent paleoclimate studies suggest the expansion and contraction of the ITCZ during the last millennium (Utida et al., 2019; Asmerom et al., 2020). However, it is challenging to assess the oscillatory nature of precipitation and the ITCZ due to the scarcity of paleorecords and the complex relationship between the position, width, and intensity of the ITCZ (Byrne et al., 2018).

The understanding of the mechanisms that regulate the position of the ITCZ is still limited. Climate models provide dynamic/thermodynamic variables that can enhance the understanding of these processes and fill an important knowledge gap. According to Ortega et al. (2021), biases in the representation of the ITCZ during the summer and spring seasons are reduced in the Coupled Model Intercomparison Project phase 6 (CMIP6) compared to the Coupled Model Intercomparison Project phase 5 (CMIP5). However, the bias of the double ITCZ over the Atlantic is only slightly reduced, and there still remains a large spread among the models in CMIP6 (Tian and Dong, 2020). Several studies support the improvement of CMIP6 models compared to CMIP5 models. CMIP6 models outperformed CMIP5 models in terms of accurate SST estimates and improved simulation of SST variability patterns in regions like the tropical Pacific and tropical Atlantic (Li et al., 2020). Additionally, CMIP6 models

exhibited significant improvements in the position of the ITCZ in the Atlantic compared with CMIP5 models, although most CMIP6 models tend to overestimate the interannual variability of the ITCZ position (Richter and Tokinaga, 2020). Taking into account these uncertainties observed in the displacement and intensity of the ITCZ, we employed a probabilistic approach proposed by Mamalakis and Foufoula-Georgiou (2018).

Thus, we provide a comprehensive and quantitative assessment of which aspects of the variability of the tropical Atlantic ITCZ are adequately represented by current models. In relation to the study of multidecadal variability, we employed a low-frequency component analysis (LFCA) applied to the averaged monthly data from the MRI-ESM2-0 (Yukimoto et al., 2019) and MIROC-ES2L (Hajima et al., 2019) models of CMIP6. The LFCA method was applied to characterize and understand low-frequency modes of SST variability in the Atlantic and Pacific oceans (Wills et al., 2018, 2019). Based on LFCA, it was determined which mechanisms are responsible for the decadal to multidecadal variability of the ITCZ, particularly over the tropical Atlantic Ocean region. The main objective of this study was to examine the decadal to multidecadal variability of the ITCZ, particularly over the tropical Atlantic Ocean region, using output from climate simulations covering the last millennium (850 – 1850 AD). Specifically, the causes of trends as well as the variability in position and intensity of the ITCZ on a multidecadal scale were examined through the analysis of outputs from the CMIP6 simulations. These results will be further verified through coherence with estimates derived from paleoclimatic proxies over South America (Vuille et al., 2012; Utida et al., 2019).

The organization of this manuscript is as follows. After this brief introduction, a description of the study area and data is presented, followed by the methods used. Next, the results of the uncertainties in the simulations by the CMIP5 and CMIP6 models, in terms of the position of the ITCZ we used a probabilistic method, and of the multidecadal variability in space and time are presented below for the simulations of the CMIP6 Models during MCA and LIA in the NEB. This article ends with a discussion followed by conclusions.

**2   Study area and data description**

**2.1   Study area**

The variability of SST in the Tropical Atlantic (TA) plays a significant role in influencing the precipitation regime, which has important implications for the populations of Africa and South America (Seager et al., 2010). In the NEB region, the SST of the TA and tropical Pacific oceans are considered the primary physical variables that contribute to climate variability conditions (Philander, 1989). In the Equatorial Atlantic (EA), the gradients of SST towards the southern latitudes have a significant influence on the total precipitation in the NEB region. These SST gradients play a crucial role in modulating the latitudinal positioning of the ITCZ, which is a key driver of rainfall patterns. Changes in the SST gradients in the EA can result in shifts in the position of the ITCZ, thereby impacting the distribution and intensity of precipitation in NEB (Hastenrath and Greischar, 1993; Hastenrath, 1984; Nobre and Shukla, 1996; Marengo, 2004; Chang et al., 2006; Marengo et al., 2017; Deser et al., 2010).

In response to the warming of SST in the Atlantic and the variability in the intensity of the trade winds, the ITCZ presents a latitudinal displacement throughout the year (Waliser and Gautier, 1993; Waliser and Jiang, 2015). The changes in SST and

trade winds can influence the atmospheric circulation patterns, leading to a movement of the ITCZ and consequently impacting the distribution of rainfall in different regions. Between February and March, the ITCZ is typically located between the equator and around 5°N in response to specific atmospheric and oceanic conditions. During this period, the South Atlantic experiences higher SST, which contributes to the northward displacement of the ITCZ. Additionally, the northeast trade winds intensify during the austral summer, further influencing the latitudinal positioning of the ITCZ. These factors combined result in the ITCZ being situated in the specified latitude range during this time of the year. Between July and August, the ITCZ tends to be located approximately between 5°N and 8°N in the western part of the TA. This positioning is influenced by several factors. First, the North Atlantic experiences higher SST during this period, contributing to the northward displacement of the ITCZ.

**2.2 Description of the dataset**

The model evaluation was based on historical simulations of CMIP5 and CMIP6 models, as shown in Table 1. The main objective was to quantitatively assess the estimation of the variability of the ITCZ in the tropical Atlantic. The first analysis compared the average annual rainfall cycles simulated by the CMIP5 and CMIP6 models with the observed pattern of the Global Precipitation Climatology Project (GPCP). In the second analysis, the modes of variability in the position of the ITCZ will be validated by comparing the smoothed series of records from the Cariaco Basin with the data from Lake Boqueirão.

**2.2.1 Model Data**

The Last Millennium (LM) experiments, designed to contribute to the Coupled Model Intercomparison Project Phase 6 (CMIP6) as described by Eyring et al. (2016), have been incorporated into the Paleoclimate Modelling Intercomparison Project Phase 4 (PMIP4). These simulations encompass the period from 850 to 1850. The models data can be downloaded from the following URL: https://esgf-node.llnl.gov/search/cmip6. We utilized the LM outputs from two models: MIROC-ES2L (Hajima et al., 2019) and MRI-ESM2-0 (Yukimoto et al., 2019). These models were chosen specifically because they were the only ones that provided data for all five climatic variables considered for the last millennium period (850 – 1850 AD). For our study, we specifically chose the first and second realizations of the MIROC-ES2L and MRI-ESM2-0 models (designated as r1i1p1f2 and r1i1p1f1, respectively). These selections were made due to the unavailability of the first realization in these models. The monthly scalar variables used in our analysis include precipitation (PR), sea surface temperature (SST), zonal and meridional wind components (u and v), and the Lagrangian tendency of air pressure ($\omega$).

To investigate the potential association with Atlantic Multidecadal Oscillation (AMO) variability, the AMO index is calculated based on the detrended time series of sea surface temperatures in the North Atlantic region (Enfield et al., 2001). The AMO index represents the variability of sea surface temperatures in the North Atlantic region (0°-80°N) and can provide insights into long-term climate variations.

**Table 1.** List of outputs from the CMIP5 and CMIP6 models used in this study.

| N° | INSTITUTION | CMIP5 | Atmospheric grid (lat.×lon.) | CMIP6 | Atmospheric grid (lat.×lon.) |
|---|---|---|---|---|---|
| 1 | Meteorological Research Institute (MRI) | MRI-ESM1 | $1.1° \times 1.1°$ | MRI-ESM2-0** | $1.1° \times 1.1°$ |
| 2 | Atmosphere and Ocean Research Institute (AORI) | MIROC-ESM | $2.8° \times 2.8°$ | MIROC-ES2L* | $2.7° \times 2.8°$ |
| 3 | National Aeronautics and Space Administration (NASA) | GISS-E2-H | $2.0° \times 2.5°$ | GISS-E2-1-G | $2.0° \times 2.5°$ |
| 4 | Met Office Hadley Centre | HadCM3 | $3.7° \times 2.5°$ | HadGEM3-GC31-LL | $1.25° \times 1.87°$ |
| 5 | Institut Pierre-Simon Laplace (IPSL) | IPSL-CM5A-LR | $1.9° \times 3.8°$ | IPSL-CM6A-LR | $1.3° \times 2.5°$ |
| 6 | Max Planck Institute for Meteorology (MPI-M) | MPI-ESM-P | $1.9° \times 1.9°$ | MPI-ESM1-2-LR | $1.87° \times 1.87°$ |
| 7 | Beijing Climate Center (BCC) and China Meteorological Administration (CMA) | BCC-CSM-1-1-m | $2.8° \times 2.8°$ | BCC-CSM2-MR | $1.1° \times 1.1°$ |
| 8 | Atmosphere and Ocean Research Institute (AORI) | MIROC5 | $1.4° \times 1.4°$ | MIROC6 | $1.4° \times 1.4°$ |
| 9 | Australian Community Climate and Earth System Simulator (ACCESS) | ACCESS1-0 | $1.3° \times 1.9°$ | ACCESS-CM2 | $1.2° \times 1.8°$ |

Note: the resolution of the historical ensemble is $2.5° \times 2.5°$. While CMIP6 averages for the last millennium have a resolution of $1.1° \times 1.1°$. Model vegetation distribution *natural and **prescribed.

**2.2.2 Observed Data**

To investigate the seasonal behavior of the ITCZ, we used the Global Precipitation Climatology Project (GPCP) dataset with a spatial resolution of 2.5°x2.5° (latitude and longitude) (Adler et al., 2003). Additionally, we obtained monthly data of Outgoing Longwave Radiation (OLR) from the National Oceanic and Atmospheric Administration (NOAA). The OLR data also have a spatial resolution of 2.5°x2.5° (latitude and longitude) and cover the period from 1981 to 2005. The analysis is based on the inverse relationship between OLR and rainfall. As noted by Liebmann and Smith (1996), higher OLR values are associated with a lower chance of rainfall, whereas lower OLR values are associated with a higher chance of rainfall.

**2.2.3 Proxy data**

In our study, we conducted a comparison between the simulations and multiproxy-reconstructed paleoclimatic records from South America. We specifically utilized proxy precipitation data that has been previously published by Vuille et al. (2012) and Utida et al. (2019). These proxy records provide valuable insights into past precipitation patterns in the region and serve as important references for validating our model simulations. The seasonal climate in coastal regions of the tropical Atlantic, such as the NEB and the Cariaco basin (located at coordinates 10.7°N, 65.16°W), is primarily influenced by the movement of the ITCZ. Studies by Utida et al. (2019) and Campos et al. (2019) have highlighted the significant role of the ITCZ displacement in shaping the seasonal climate patterns in these areas. The *Boqueirão* lake is situated near the eastern coastline of the NEB region (located at coordinates 5.23°S, 35.53°W). It plays a crucial role in the water supply as approximately 50% of the precipitation in this area occurs during the months of March, April, and May (MAM) when ITCZ is located further south (Utida et al., 2019). During the austral winter months of June, July, and August (JJA), notable climate features are observed in the Tropical South Atlantic (TSA). The TSA experiences significant cooling during this period, which is accompanied by the strengthening of southeast trade winds that traverse the Equator. Additionally, the ITCZ undergoes a northward displacement. These changes in the climate patterns during JJA have been documented in the study by Utida et al. (2019).

The hydrogen isotope composition of the n-C28 alkanoic acid ($\delta$D) extracted from *Boqueirão* lake (Boqc0901 core), as studied by Viana et al. (2014), provides valuable information about past climate conditions. Variations in the $\delta$D values can be used as a proxy for changes in precipitation associated with the ITCZ. Lower $\delta$D values indicate increased ITCZ-related precipitation, while higher $\delta$D values reflect reduced precipitation. By analyzing the $\delta$D record, Viana et al. (2014) were able to reconstruct past changes in ITCZ activity and its influence on regional precipitation. The Cariaco Basin sediment record, as studied by Haug et al. (2001), provides insights into past changes in rainfall and the position of the ITCZ. The study found that a decrease in titanium (Ti) concentration in the Cariaco Basin sediment indicates reduced rainfall and increased aridity in the region. On the other hand, Ti peaks in the sediment record suggest that the ITCZ has shifted to a more northern position.

**3 Methods**

**3.1 Low-frequency component analysis (LFCA)**

The LFCA is a method used to identify patterns with a maximum ratio of low-frequency to total variance in a dataset. It achieves this by transforming the leading Empirical Orthogonal Functions (EOFs) of the data and applying a low-pass filter. The resulting patterns, known as Low-Frequency Patterns (LFPs), and their corresponding components, called Low-Frequency Components (LFCs), represent the low-frequency variability in the data. The LFCA method, as described by Wills et al. (2018), enables the disentanglement of low-frequency signals from higher-frequency noise in a dataset. By isolating the low-frequency patterns, we can gain insights into the underlying dynamics and variability of the studied system on longer timescales. The LFCA technique can be valuable for understanding and analyzing climatic, as it helps identify and separate low-frequency variability patterns that may be associated with significant climate modes or long-term trends.

To analyze the variability of the ITCZ associated with the LIA and the MCA, the LFCA was applied to SST and precipitation anomalies in the region between 30°S and 30°N. Additionally, the SST in the tropical Atlantic region was included in the analysis. The LFCA method was used to identify patterns with the highest ratio of low-frequency to total variance in the long-term simulations of SST and precipitation anomalies for the last millennium. Specifically, the monthly data of the ensemble, averaging the models MRI-ESM2-0 and MIROC-ES2L. By applying LFCA to the SST and precipitation anomalies, we aimed to uncover the low-frequency variability patterns associated with the ITCZ during the LIA and MCA periods. This analysis provides insights into how the ITCZ position and precipitation patterns varied during these climatic epochs.

**3.2 Hodrick-Prescott smoothing (HP)**

The Hodrick-Prescott (HP) is a linear filter that requires the prior specification of a parameter. In joint analysis, this parameter should be the same for filtering all series. A higher parameter value leads to a smoother resulting series, while a lower value allows for more variability (Hodrick and Prescott, 1997). The HP filter is commonly used to separate a time series into its trend and cyclical components. By applying the filter, the long-term trend can be isolated from the short-term fluctuations, enabling the analysis of underlying patterns and variations in the data.

The value 14400 is commonly used when applying the HP filter to monthly time series (Maravall and Del Rio, 2001). This value is chosen to effectively filter out components in the time series with periods of fewer than 10 years and interdecadal cycles (Vásquez et al., 2018). By setting the parameter of the HP filter to 14400, the resulting filtered series will have a smoother trend and remove shorter-term fluctuations, allowing for a clearer analysis of long-term patterns and variations in the data. This specific value has been found to be suitable for capturing medium to long-term trends and suppressing high-frequency noise in monthly time series analyses. The HP filter was applied to both the paleoclimatic records and the LFCs precipitation from the CMIP6 models. The purpose of applying the HP filter was to uncover any potential trends, phase synchrony, and temporal variability in the precipitation data and paleoclimatic records.

**3.3 Probabilistic tracking of the ITCZ**

We utilized the approach proposed by Mamalakis and Foufoula-Georgiou (2018) to estimate the probability density function (PDF) for rainfall. This method allowed for a comprehensive analysis of the seasonal variations in the ITCZ and offered the flexibility of using multiple variables to define the ITCZ, enhancing its physical robustness. By estimating the PDF for rainfall, we gained insights into the statistical distribution of rainfall values and their variations across different seasons and periods. Moreover, the PDF analysis method is computationally efficient and adaptable, making it well-suited for analyzing multi-model ensembles in climate change assessment studies. In our study, we applied the PDF analysis to investigate the climatology of the ITCZ, with a specific focus on its seasonal and annual average position along the longitudinal axis. Our primary objective was to assess the ability of the CMIP5 and CMIP6 models (as listed in Table 1) to replicate the inter-annual and intra-seasonal variability of precipitation associated with the ITCZ

**3.4 Red-noise model and statistical significance**

The univariate spectra were bias-corrected using 1000 Monte Carlo simulations. This was achieved using the publicly available REDFIT software (Schulz and Stattegger, 1997), which generates first-order autoregressive (AR1) time series with sampling times and characteristic timescales matching those of the actual climate data. The REDFIT software can be found at the following file location: /pub/sfb313/mschulz/redfit35.zip. We employed this methodology to analyze the LFC components of precipitation in the CMIP6 models and their potential association with AMO variability and its phases. The spectra were color-coded based on the age of the time window's midpoint and highlighted frequencies where the spectral power exceeded the red-noise false-alarm levels (confidence levels) of 95%.

**4 Results and discussion**

**4.1 Latitudinal Position of the ITCZ**

In order to evaluate and identify the uncertainty of the position of the ITCZ in the Tropical Atlantic in the outputs of the CMIP models (Table 1), we will analyze the seasonal (DJF, MAM, JJA, and SON) and interannual position during the period of 1981-

[Figure]

**Figure 1.** The probability density function (PDF) represents the distribution of the ITCZ location on seasonal scales. To calculate the PDF, we utilize the joint statistics of observed window-mean precipitation and outgoing longwave radiation (OLR).

2005. The probabilistic method proposed by Mamalakis and Foufoula-Georgiou (2018) was used to obtain the spatial mean of the probability distribution of the ITCZ position over the Tropical Atlantic. This method utilizes two meteorological variables: precipitation (PR) and outgoing longwave radiation (OLR). These results were compared with the estimated position of the ITCZ during the last millennium (850–1849) using the ensemble outputs of the MRI and MIROC. For the period of 1981–2005, during the austral summer and autumn seasons (DJF and MAM), our results indicate that the CMIP models estimate the position of the ITCZ with a pronounced southward displacement. Figure 1 shows a seasonal bias in the position of the ITCZ compared to the observed data, with a larger discrepancy observed during the DJF and MAM seasons. For all seasons, the results from GPCP indicate a predominantly northern position of the ITCZ (Figure 1c). This finding contrasts with recent studies that highlight the limitations of CMIP models in simulating the position and intensity of the ITCZ (Richter and Xie, 2008; Richter et al., 2012; Zermeño-Diaz and Zhang, 2013; Shonk et al., 2019; Mamalakis et al., 2021). These model biases may be associated with the parameterization of atmospheric and oceanic processes (Zhang et al., 2019; Song and Zhang, 2020). Mamalakis et al. (2021) found that CMIP models underestimate the interannual variability of the ITCZ position in the tropical Atlantic, suggesting difficulties in capturing the dynamic processes that influence the ITCZ position in this region.

[Figure]

**Figure 2.** The probability density function (PDF) represents the distribution of the ITCZ location on an annual scale.

Regarding annual variability, the results show that the highest probability for the location of the ITCZ is between $0°$ and $10°N$ for both CMIP models and the observed data (Figure 2). Mamalakis and Foufoula-Georgiou (2018) observed that the ITCZ position in the Atlantic remains predominantly in the Northern Hemisphere throughout most of the year. They also noted that CMIP models tend to underestimate the relationship between the ITCZ position and SST in the tropical Atlantic, suggesting that these models may overestimate the influence of SST on the modulation of the ITCZ position. The authors further emphasize the possibility of a future southward shift of the ITCZ, which would be associated with the models' overestimation of SST in the equatorial basin region.

In Figure 2, it can be observed that the CMIP models display a second band of probability, which is located between $0°$ and $10°S$. Several studies have noted that the presence of the double ITCZ is reduced when modifications are made to the parameterization scheme of convection (Song and Zhang, 2009; Chikira and Sugiyama, 2010; Oueslati and Bellon, 2013, 2015; Song and Zhang, 2018). Conversely, Lu et al. (2021) demonstrated that the bias can be mitigated by modifying the parameterization scheme of boundary layer turbulence and shallow convection. The issue of the double ITCZ in models has also been studied by Hwang and Frierson (2013), who highlighted that it stems from an energy balance bias between the two hemispheres. They observed that excess energy absorbed in the southern hemisphere is transported to the northern hemisphere through the upper branch of the anomalous Hadley cell over the equator, while the lower branch transports water vapor southward, leading to a southward displacement of the ITCZ and the formation of the double ITCZ. Additionally, Adam et al. (2016) suggests that CMIP5 models exhibit a positive bias in atmospheric energy transport, which can contribute to a shift in the formation of the double ITCZ. This bias may be associated with errors in the representation of physical processes and atmospheric interactions in the models, resulting in an inadequate simulation of the ITCZ and its variability patterns. Overall, the models tend to exhibit a bias over the tropical Atlantic, overestimating the probability of the ITCZ migrating southward (Figure 1). Tian and Dong (2020) observed that annual mean precipitation simulations in CMIP3, CMIP5, and CMIP6 models exhibit similar systematic errors, indicating that the bias of the double ITCZ remains a persistent problem across all generations of climate models. They highlighted a slight reduction in the double ITCZ bias over the Pacific and Atlantic oceans from CMIP5 to CMIP6.

Figure 3 illustrates the normalized difference in the PDF between the MCA and LIA periods for both CMIP5 and CMIP6. The figure shows a meridional gradient, indicating an increased probability towards the south in both cases. This suggests that

[Figure]

**Figure 3.** The normalized difference in the probability density function (PDF) between the Medieval Climate Anomaly (MCA) and Little Ice Age (LIA) for (a) CMIP5 (MRI-ESM1 and MIROC5) and (b) CMIP6 (MRI-ESM2-0 and MIROC-ES2L).

while CMIP models can capture the displacement of the ITCZ, they tend to underestimate the meridional variability, especially towards the north. During the MCA, there is no clear position of the ITCZ over the Northern Hemisphere, resulting in little difference between the MCA and LIA in terms of the ITCZ position. Additionally, the limited variability in the displacement of the ITCZ during the MCA may be attributed to its representation, as mostly uniform PDF values are observed, indicating the presence of a possibly very homogeneous cloud band. This limited representation may hinder the capture of the meridional variability of the ITCZ. Although both CMIP models do not exhibit the expected behavior of the ITCZ during the MCA, it is observed that CMIP6 shows PDF values with a greater tendency north of the equator compared to CMIP5. The CMIP6 exhibits a more defined PDF gradient and is able to identify the part of the South Atlantic Convergence Zone (SACZ). These findings indicate a notable improvement in the results of the CMIP6 model, which is consistent with the findings of Richter and Tokinaga (2020).

Regarding the variability of the ITCZ displacement during the MCA and LIA, both models indicate that the ITCZ position was further south during the LIA compared to the MCA. However, the CMIP6 model shows a higher probability of accurately representing the intensity and displacement of the ITCZ compared to CMIP5 (Figure 3). Oxygen isotope data (obtained from NEB), supports the characterization of the MCA as a dry period (Novello et al., 2012), influenced by the intensity of the South American Monsoon System (SAMS) and internal forcing. However, paleolimnological data suggests a wet period in the northern region of the NEB during the MCA. This is evidenced by the increase in water level in the *Boqueirão* lake (Viana et al., 2014) and the northward displacement of the ITCZ, with wet conditions recorded in the Cariaco Basin (Haug et al., 2003). On the other hand, the LIA was characterized by a southward migration of the ITCZ (Vuille et al., 2012), which corresponded to drought conditions in the Cariaco Basin (Haug et al., 2003).

The CMIP5 and CMIP6 models exhibit a similar spatial distribution of the PDF for MCA and LIA. In both cases, the prevailing position of the ITCZ is observed to be south of the equator. These findings differ from various studies based on paleohydrological records, which indicate that during the MCA, the dominant position of the ITCZ is north of the equator

(Kleppe et al., 2011; Vuille et al., 2012). This discrepancy highlights one of the main shortcomings of the CMIP5 and CMIP6 models in accurately representing the ITCZ.

**4.2 The Interdecadal Component of the ITCZ latitudinal location**

To investigate the multidecadal variability of the ITCZ in the southern tropical Atlantic, we conducted a Low-Frequency Component Analysis (LFCA) using the CMIP6 models MRI-ESM2-0 and MIROC-ES2L. This analysis allowed us to isolate the long-term precipitation variability and identify two modes of ITCZ displacement. The first mode corresponds to a southward displacement, represented by precipitation LFC components 1 and 3, while the second mode indicates a northward displacement, represented by LFC components 2 and 4. Figure 4 presents the Cariaco Basin and *Boqueirão* Lake data, which, due to their strategic locations, provide insights into the latitudinal variations of the ITCZ with an antiphase relationship.

In Figure 4a, a predominant presence of positive precipitation values can be observed in the LFC components (2 and 4) during the MCA. These results are consistent with the records from the Cariaco Basin, where a higher concentration of titanium during the MCA has been observed. This indicates that the ITCZ is shifted northward, leading to humid conditions. On the other hand, during the LIA, a higher presence of negative values is observed in the LFC components (2 and 4), contrasting with the decrease in titanium concentration. These records indicate dry conditions during that period (Haug et al., 2003). Furthermore, this observed condition is associated with the southward migration of the ITCZ (Haug et al., 2001; Bird et al., 2011; Vuille et al., 2012; Novello et al., 2012). In summary, the modes of variability represented by components 2 and 4, simulated by the CMIP6 models, reflect the variability of the ITCZ position.

Regarding Figure 4b, a predominant presence of positive precipitation values can be observed in the LFC components (1 and 3) during the MCA, which differs from the data from Lake Boqueirão ($\delta$D). Positive $\delta$D values indicate dry periods in the NEB region, associated with the northward displacement of the ITCZ (Novello et al., 2012). However, components 1 and 3 do not consistently indicate dry conditions as they do not predominantly record negative values.

On the other hand, during the LIA, the LFC components (1 and 3) exhibit fluctuations between negative and positive values. Negative values of $\delta$D data indicate wet periods in the northern region of the NEB, as supported by the increased water level of Lake Boqueirão (Viana et al., 2014). This is associated with the southward position of the ITCZ. However, this variability in components 1 and 3 could be related to what was observed by Novello et al. (2012), who suggest that the intensification of the Northeast High was responsible for the drought in the southern NEB during an intense phase of the SAMS.

**4.3 Wavelet coherence analysis**

To better understand the dynamic link between climate and dominant modes of variability, we applied wavelet coherence analysis to the low-frequency component anomalies of precipitation over the last millennium (Figure 5). This analysis reveals the interdecadal and multidecadal variability, as well as the periods that could be associated with AMO.

In Figure 5, a strong signal coherence is observed between the LFC and the AMO signal, displaying periodicities ranging from 2 to 8 years with a coherence coefficient of 0.9. It is important to note that this periodicity appears intermittently over the analyzed period. This interannual variability is thought to be linked to the oscillations between the warm and cold phases

[Figure]

**Figure 4.** HP filter applied to time series $(\mathrm{mm/year})$ with $\lambda = 14400$ of paleoclimatic proxies and precipitation anomalies of low-frequency components (LFC) throughout the last millennium in the CMIP6 ensemble (average of MRI-ESM2-0 and MIROC-ES2L) based on low-frequency component analysis with 30 retained EOFs and a 10-year low-pass cutoff. (a) Ti Cariaco basin ((blue line)) (Haug et al., 2001), LFC2 and LFC4 (grey lines). (b) $\delta$Dwax *Boqueirão* Lake (blue line) (Viana et al., 2014), LFC1 and LFC3 (grey lines).

of the AMO, which subsequently influence the occurrence of El Niño and La Niña events. These variations in oceanic and atmospheric conditions can influence the precipitation patterns over the NEB region, leading to periods of wetter or drier conditions.

Additionally, Figure 5 also reveals a notable 24-year periodicity between 949 and 1049, where the LFC1 signals and the AMO exhibit an antiphase relationship of 180°. On the 64-year scale (1399 to 1549), the variables have a phase angle of 90°, which means that the maximum precipitation components occur 16 years after the maximum of the AMO. In Figure 5b, a periodicity of 64 years is observed between the years 949 and 1249, where the variables of LFC2 and the AMO have a phase angle of 45°, with maximum precipitation occurring 8 years after the maximum of the AMO. For the 80-year period, both signals are in phase (180°), indicating that they vary coincidentally, reaching their maximum or minimum values at the same time.

For the LFC3 component and the AMO (Figure 5c), a periodicity of 24 years is observed between the years 1249 and 1349, where the maximum of the AMO occurs 6 years after the maximum of precipitation. For the 64-year period between the years 950 and 1049, the signals exhibit two phases. In the first phase, with an angle of 135°, the maximum precipitation occurs 24

[Figure]

**Figure 5.** (a) The wavelet coherence of AMO indices during the last millennium in the ensemble (averaging the models MRI-ESM2-0 and MIROC-ES2L) is shown for the following components: (a) LFC1, (b) LFC2, (c) LFC3, and (d) LFC4, representing low-frequency patterns of precipitation. The wavelet power spectra are represented by thick black contours, enclosing areas of 5% significance against a red-noise background. The cone of influence is represented by light shading. In the global power spectra, the thick black lines represent the global wavelet power estimates, and the thin dashed black lines represent the 95% confidence bounds against the red-noise background spectra.

years after the maximum of the AMO, while in the second phase, with an angle of 90°, the maximum precipitation occurs 16 years after the maximum of the AMO. For the periodicity of 128 years between the years 1549 and 1650, the variables are related with an angle of 90°, with the maximum precipitation occurring 32 years after the maximum of the AMO.

For LFC4 (Figure 5d), the multidecadal variability of 12 years is present during the period of 1049 to 1060 and 1265 and 1460. The multidecadal periodicities of 21 and 128 years, during the periods 1249 to 1270, indicate that the signals are out of phase. For the 80-year period from 1450 to 1550, the LFC4 signal and the AMO are in antiphase (180°), meaning that the maximum precipitation coincides with the minimum of the AMO.

The significant periodicities observed during the last millennium in Figure 5 (e and f) are in the range of 2 to 8 years and 2 to 4 years, respectively. These periodicities are associated with internal variability in the changes of the ITCZ position and may be influenced by ENSO events. However, there are differing views on the variability of ENSO during the last millennium. Some studies suggest a lower frequency of El Niño during the MCA, while others argue for a higher frequency of El Niño during

the LIA, possibly related to increased volcanic activity during that period Jones and Mann (2004); Mann and Emanuel (2006); McGregor and Timmermann (2011); Predybaylo et al. (2017); Henke et al. (2017). The average power spectrum (Figure 5e) illustrates the signals from the LFC precipitation components (1 and 3) and their relationship with the meridional displacement of the ITCZ towards the south (Figures 6 and 8). The presence of multidecadal periodicities between the LFC3 signal and the AMO can be observed, particularly during the transition period between the MCA and the LIA. These periodicities are believed to be influenced by volcanic forcing, which played a significant role in explaining the low solar radiation activity during the LIA (Miller et al., 2012).

Indeed, the southward displacement of the ITCZ during the LIA is considered a response to the relative cooling of the Northern Hemisphere driven by volcanic forcing. Volcanic eruptions release large amounts of aerosols and gases into the atmosphere, which can lead to a cooling effect by scattering and absorbing sunlight (Mann et al., 2021). This cooling effect is particularly pronounced in the Northern Hemisphere, where most volcanic activity occurs. The reduced solar radiation reaching the Earth's surface results in a cooling of the Northern Hemisphere, altering the temperature gradients and atmospheric circulation patterns. As a result, the ITCZ shifts southward, reflecting the changes in the distribution of solar energy and temperature gradients across latitudes. This southward displacement of the ITCZ during the LIA is a manifestation of the complex interactions between volcanic forcing, hemispheric temperature anomalies, and atmospheric circulation patterns.

Sun et al. (2022) discovered a significant impact of volcanic activity during the LIA on El Niño variability. They found that periods of intense volcanic activity in the Northern Hemisphere were associated with an increased frequency of El Niño events. This relationship suggests that volcanic activity can play a role in short-term climate variability, contrasting with its long-term influence on multidecadal variability. Additionally, notable cycles in the Pacific Decadal Oscillation (PDO) were observed during the MCA and LIA. Both periods exhibited an intrinsic cycle of 20-40 years in the PDO. During the MCA, a cycle of approximately 70 years, known as the Gleissberg cycle, was identified, which is associated with solar activity variability. On the other hand, during the LIA, a more pronounced forced cycle of 50-70 years was observed, which has been linked to oceanic circulation variability, such as the AMO. Levine et al. (2018) further investigated the relationship between the AMO and the variability of the ITCZ. They found that during positive phases of the AMO, the ITCZ shifted towards the warmer hemisphere, resulting in increased precipitation over the North Atlantic. This indicates that AMO phases are associated with ITCZ variability through surface variability and energy transport between hemispheres. The study also highlighted that during positive AMO phases, the interannual variability of the tropical Atlantic ITCZ is greater.

The average power spectrum depicted in Figure 5f illustrates the signals of the LFC precipitation components (2 and 4) and their association with the northward displacement of the ITCZ as shown in Figures 7 and 9. This variability is believed to be linked to the positive phase of the AMO. When the North Atlantic experiences warmer conditions, it triggers a change in the interhemispheric gradient, resulting in the northward shift of the ITCZ (Knight et al., 2006; Timmermann et al., 2007). According to Levine et al. (2018), the significant periodicity of 80 years observed in Figure 5 is attributed to the 50-80 year variability found in the AMO. This variability is linked to a precipitation band that stretches along the ITCZ from the tropical Indian Ocean and western Pacific to the tropical Atlantic. The presence of this precipitation band is associated with a coherent zonal structure in atmospheric variability in mid and high latitudes (Lin et al., 2019). Indeed, external forcing, including

radiative forcing from volcanic eruptions, can play a significant role in influencing variables such as the AMO and its associated multidecadal oscillatory signals. Mann et al. (2021) observed similar multidecadal oscillatory signals to the AMO in last millennium simulations, specifically in the 50-70 year band, and they determined that these signals were exclusively driven by radiative forcing resulting from volcanic eruptions. These findings are important because they suggest that AMO phases can significantly affect the position and variability of the ITCZ in the tropical Atlantic, which in turn can have implications for precipitation and temperature in regions near the equator.

The interannual variability of the AMO and the ITCZ plays a crucial role in understanding natural climate variability and its potential future changes. The results demonstrate that during the last millennium, changes in the position of the ITCZ have comparable significance to external forcing factors. The influence of internal forcing on the ITCZ changes in the tropical Atlantic is characterized by modes of variability such as ENSO, PDO, and AMO, which exhibit spatial patterns showing latitudinal displacement between northern and southern regions. The ENSO phenomenon, for example, can induce changes in the atmospheric circulation patterns and sea surface temperatures, which in turn can impact the position of the ITCZ. Similarly, the PDO and AMO modes can influence large-scale atmospheric and oceanic conditions, affecting the latitudinal displacement of the ITCZ. These findings highlight the importance of comprehending the dynamics and interactions between the AMO and the ITCZ. By understanding the behavior of these internal modes of variability, we can gain insights into regional climate variability and make more accurate projections for future climate conditions.

**4.4 The Interdecadal Component of the ITCZ latitudinal location**

Figure 6a presents the first mode (LFP1) of SST variability. It shows an increase in the meridional gradient of the Atlantic associated with the weakening of winds, which reduced the surface latent heat flux, leading to increased SST. It also shows a southward displacement of the ITCZ, with its center of action located approximately at $5°S$ and between longitudes $20°W$ and $0°W$ (Figure 6b). In addition, the SACZ extends its influence into the Atlantic Ocean, adjacent to the Southeast region of Brazil. This is associated with a periodicity in the range of 2 to 8 years, resulting from the internal variability of ITCZ changes, which are linked to La Niña events and intensified precipitation in the NEB. These findings are consistent with other studies (Grimm, 2003, 2004; Grimm and Tedeschi, 2009). Negative values of Omega are observed in front of the NEB (Figure 6c), which are associated with upward motions and favor the occurrence of precipitation in the NEB and its vicinity. The winds at $850\,\mathrm{hPa}$ on the southern side of the Atlantic predominantly blow westward, leading to the accumulation of warmer waters along the African coast (Figure 6a). This implies a reduction in the upwelling process along these coasts.

In Figure 7c, the 850 hPa winds exhibit a similar pattern to Figure 6c, but with the addition of eastward winds in the westernmost part of the South Atlantic. This leads to a more pronounced upwelling process compared to Figure 6a. Furthermore, Figure 7a shows a positive core of SST north of $20°N$, and the ITCZ extends further meridionally compared to its climatological value. These conditions contribute to the humid conditions observed in the regions where the proxies were extracted, namely *Boqueirão* and Cariaco.

The eastern region of the Amazon exhibits wetter conditions, while a precipitation deficit is observed in the southern Amazon (Figure 7b). This suggests that the precipitation patterns of the SAMS feature wetter conditions in its northeastern part and drier

[Figure]

**Figure 6.** The first component of low-frequency patterns (LFP1) represents (a) the sea surface temperature anomalies (°C), (b) precipitation anomalies (mm/month), and (c) 500 hPa omega vertical motion and wind at 850 hPa throughout the last millennium. These analyses were conducted in the ensemble by averaging the models MRI-ESM2-0 and MIROC-ES2L. The low-frequency component analysis involved retaining 30 EOF and applying a 10-year low-pass cutoff. The location of the paleoclimate proxies are (1) *Boqueirão* Lake (Viana et al., 2014) and (2) Cariaco Basin (Haug et al., 2001).

[Figure]

**Figure 7.** Same as figure 6 but for the second component of Low-frequency patterns (LFP2).

conditions in its southern and southwestern extremes. This pattern of precipitation distribution is associated with the positive phase of AMO, which has a periodicity of 80 years. According to Levine et al. (2018), the latitudinal changes of the ITCZ and the strength of the AMOC could be related through the surface variability of the SST, exhibiting similar characteristics of AMO variability. The authors suggest that during the positive phase of the AMO, the ITCZ shifts towards the warmer hemisphere, resulting in increased precipitation in the North Atlantic. If the warming is more pronounced in the Northern Hemisphere, the ITCZ tends to move further north. An existing relationship between the ITCZ and the atmospheric energy balance can account for the variability of the ITCZ on timescales ranging from years to geological scales (Schneider et al., 2014; McGee et al., 2014; Schneider, 2017). Some studies show that the positive phase of the AMO can lead to a weakening of the SAMS intensity on multidecadal timescales, due to a northward shift of the ITCZ (Stríkis et al., 2011; Bird et al., 2011; Novello et al., 2012). Moreover, this displacement of the ITCZ results in a decrease in precipitation over the Northeast region of Brazil (Knight et al., 2006).

[Figure]

**Figure 8.** Same as figure 6 but for the third component of Low-frequency patterns (LFP3).

In Figure 8, it can be observed that the trade winds weaken (divergence) and the southeast winds strengthen (convergence), and there is a reduction in upwelling off the northwest coast of Africa and an increased temperature gradient in the North Atlantic. These changes are all associated with the southward displacement of the ITCZ. We can also observe the presence of a double ITCZ band due to the energy balance bias between the two hemispheres. The northward displacement of precipitation in the NEB intensifies the oceanic SACZ. These precipitation patterns are associated with two significant periodicities of 24 years related to the PDO and 64 years related to the AMO. These two modes of oceanic variability are related to the intensity of the oceanic SACZ (Novello et al., 2012; Apaéstegui et al., 2014; Vuille et al., 2012; Carvalho and Cavalcanti, 2016). The multidecadal variability is related to the southward displacement of the ITCZ controlled by the inter-hemispheric heat transport (Marshall et al., 2014; McGee et al., 2014, 2018; Schneider, 2017; Bischoff and Schneider, 2014), which is modified by variations in the oceanic heat transport exerted by the AMOC. Additionally, the 24-year periodicity establishes a connection between the variability of the AMOC, the AMV, and the ITCZ. This AMV contrast is observed in the ITCZ shift following a change in the PDO (Moreno-Chamarro et al., 2019).

Figure 9c shows a wind behavior very similar to that observed in Figure 8c. While descending vertical motions are observed in the vicinity of the equator, with a more intense band located at 6°S. The strongest downward vertical motion leads to precipitation deficiency on the southern side of the Atlantic (Figure 9c). Weaker trade winds result in an SST increase in much of the northern and southern Atlantic (Figure 9a). The ITCZ band shows a position closer to its climatology (Figure 9b), but this ITCZ position increases moisture convergence from the ocean towards the continent, generating wetter conditions in the Amazon basin, intensifying the SAMS, and creating dry conditions in the NEB. These precipitation patterns are associated with two significant periodicities: a 21-year period related to solar activity and an 80-year period linked to the positive phase of the AMO. Another factor that intensifies the SAMS is a response to radiative forcing caused by volcanic eruptions (Stríkis et al., 2015).

In LFP1, Figures 10a and 10b show that during the LIA, the central-east region of Brazil experiences wetter conditions compared to the MCA. This wetter condition is associated with a strengthening of the SAMS, where the South Atlantic region is relatively warmer compared to the North Atlantic (Figures 11a and 11b). Conversely, For this component, during the MCA,

[Figure]

**Figure 9.** Same as figure 6 but for the fourth component of Low-frequency patterns (LFP4).

the Northeast Brazil (NEB) region is wetter, suggesting a strengthening of the ITCZ during that period. Additionally, it can be observed that the SACZ band extends to the Atlantic Ocean region adjacent to southeastern Brazil (Figures 10a and b). This precipitation distribution pattern was observed by Viana et al. (2014), who suggest that during the MCA, the *Boqueirão* lake was wet due to the southward displacement of the ITCZ during the negative phase of the AMO. In contrast, during the LIA, dry periods occurred due to the intensification of the Northeast High during an intense period of the SAMS.

In LFP 2 (Figure 10c), wet conditions are observed in the Cariaco Basin and dry conditions in the *Boqueirão* lake during the MCA, despite lower SST. This differs from the expected conditions for the MCA in the northern hemisphere (Figure 11c). This pattern was observed by Novello et al. (2012), who described arid conditions in the southern part of the NEB, attributed to the northward displacement of the ITCZ and the weakening of the SACZ. During the LIA, higher humidity is observed in *Boqueirão* with higher SST in the tropical South Atlantic (Figure 11d), and southward migration of the ITCZ is recorded (Figure 10d). This pattern was also shown by Haug et al. (2001); Peterson and Haug (2006); Vuille et al. (2012); Lechleitner et al. (2017). In LFP 3, during the MCA, precipitation is distributed along the South American Atlantic coast, and both the Cariaco Basin and the *Boqueirão* lake exhibit wet conditions (Figures 10e and f). Additionally, SST is warmer throughout the tropical South Atlantic (Figures 11e and f). On the other hand, during the LIA, the easternmost part of the NEB is wetter (Figure 10f), and precipitation is observed along the Amazonian coast. The SST variation between the northern and southern Atlantic is smaller, but it remains warmer in the south (Figures 11e and f). In LFP 4, an increase in precipitation is observed in the northernmost part of the NEB during the LIA period compared to the MCA (Figures 10h and g). However, in both periods, the tropical North Atlantic exhibits higher SST conditions (Figures 11g and h). The Cariaco area shows wet conditions during the LIA, which contradicts the observations made by paleoclimatic proxies (Haug et al., 2001).

The precipitation distribution patterns (LFP3 and LFP4) are sensitive to changes in SST variations in the tropical regions of the Atlantic. SST anomalies significantly impact the position and intensity of the ITCZ and can modulate the seasonal distribution of precipitation over the equatorial Atlantic, from the northern part of the NEB to the central Amazonia (Moura and Shukla, 1981; Uvo, 1989; Servain, 1991; Hastenrath and Greischar, 1993; Molion, 1993; Nobre and Shukla, 1996; Marengo, 2004; Marengo et al., 2017; Enfield and Mayer, 1997).

[Figure]

**Figure 10.** Low-frequency patterns (LFPs) precipitation anomalies $\mathrm{mm/month}$ through the MCA (850 – 1200) and LIA (1500 – 1850) in the ensemble (Averaging the models MRI-ESM2-0 and MIROC-ES2L) over the latitudes $30°$S to $30°$N, based on low-frequency component analysis with 30 EOFs retained and a 10-year low-pass cutoff. The location of the paleoclimate proxies: (1) *Boqueirão* Lake (Viana et al., 2014); (2) Cariaco Basin (Haug et al., 2001).

The meridional variations of the ITCZ play a significant role in explaining the drought conditions in the southern part of NEB during the MCA and the opposite conditions observed in the LIA. However, it is important to note that during the LIA, these conditions in the southern and northern parts of the NEB could not be solely attributed to the migration of the ITCZ. Other factors and mechanisms may have contributed to the observed changes in precipitation patterns during that period.

According to the findings of Novello et al. (2012), the intensification of the NEB High, a high-pressure system, was proposed as a factor contributing to the drought conditions in the southern part of the NEB during an intense phase of the SAMS. The NEB High, characterized by sinking air and atmospheric stability, inhibits the formation of convective systems and reduces precipitation. The intensified NEB High during the intense phase of the SAMS could have suppressed rainfall in the southern

[Figure]

**Figure 11.** Low-frequency patterns (LFPs) sea surface temperature anomalies (°C) through the MCA (850 – 1200) and LIA (1500 – 1850) in the ensemble (Averaging the CMIP6 models MRI-ESM2-0 and MIROC-ES2L) over the latitudes 30°S to 30°N, based on low-frequency component analysis with 30 EOFs retained and a 10 years low-pass cutoff. The location of the paleoclimate proxies: (1) *Boqueirão* Lake (Viana et al., 2014); (2) Cariaco Basin (Haug et al., 2001).

NEB region, exacerbating the drought conditions. Recent studies on climatology have provided evidence of a relationship between the southward shift of the SACZ and the westward shift of the Bolivian High, which subsequently displaces the NEB High over the southern NEB during the summer season. This displacement of the NEB High contributes to drought conditions in the southern NEB region (Chaves and Cavalcanti, 2001; Sulca and Rocha, 2021).

Our findings support a potential connection between paleohydrology in the NEB region and climate variations associated with fluctuations in the AMO. Moreover, we have identified that these AMO changes have the potential to influence the average position of the ITCZ. The observed meridional variations in the ITCZ provide an explanation for the occurrence of drought conditions in both the southern and northern parts of the NEB region. This displacement of the ITCZ would have weakened

the SAMS, as the reduced moisture supply promoted by it would have also reduced convection in the Amazon basin. During the MCA, precipitation is primarily distributed along the South American Atlantic coast. However, during the LIA, there is an observed increase in precipitation over the NEB region, as well as along the Amazon coast. These patterns indicate a strengthening of the ITCZ and the presence of a SACZ band that extends into the adjacent Atlantic Ocean, southeast of Brazil. The observation is that during the LIA, central Brazil experienced wetter conditions compared to the MCA. This wetter climate can be attributed to two factors: the strengthening of the monsoon and a warmer South Atlantic relative to the North Atlantic. Furthermore, the significant periodicities in the range of 2 to 8 years are associated with the internal variability of the ITCZ position changes. These periodicities are believed to be influenced by ENSO events. During the MCA, there was a lower frequency of El Niño events, whereas, during the LIA, there was a higher frequency of El Niño occurrences. This El Niño variability was associated with the meridional displacement of the ITCZ towards the south, which exhibited multi-decadal periodicities. These periodicities were found to be related to volcanic activity during that time period. The results reveal a clear pattern of variability that is linked to the phases of the AMO. Specifically, during the positive phase, there is a noticeable northward displacement of the ITCZ. In contrast, during the negative phase of the AMO, the ITCZ exhibits a southward displacement. This suggests a strong relationship between the AMO and the meridional position of the ITCZ.

**5    Conclusions**

Our results indicate that both the CMIP5 and CMIP6 models frequently simulate a southward migration of the ITCZ that is more pronounced than what is observed. The ITCZ structure in the models is strongly influenced by the seasonal cycle of precipitation, the increase in rainfall was observed to be a factor in the growth of the southern bias. The southward shift of the ITCZ in the models was observed in previous studies (Richter and Xie, 2008; Richter et al., 2014; Zermeño-Diaz and Zhang, 2013; Mamalakis et al., 2021), which causes an overestimation of the accumulated precipitation in Northeastern Brazil (NEB).

The CMIP6 and CMIP5 models tend to reproduce a double-ITCZ in the Atlantic Ocean (Figure 2). This finding contrasts with recent studies (Samanta et al., 2019; Tian and Dong, 2020; Mamalakis et al., 2021). Taking into account the reported biases, the models were able to simulate the seasonal variation of the ITCZ and reproduce the dominant modes of variability in the position in the tropical Atlantic.

On interannual to decadal timescales, the coupling between the atmosphere and the ocean plays a crucial role. Our findings indicate that the north-south displacement of the ITCZ is strongly related to the oceanic region exhibiting the highest SST within the South Atlantic tropical basin. The variability of the zonal mode is mainly associated with the equatorial region, extending between 5°S and 5°N, as well as the northwest coast of Africa. These observations also contrast with paleoclimatic records of the region, indicating a northward displacement of the ITCZ during the MCA and a southward displacement during the LIA. During the LIA, the southward displacement of the ITCZ is a response to the relative cooling of the Northern Hemisphere due to volcanic forcing. This highlights the complex interaction between solar forcing and atmospheric dynamics in shaping the behavior of the ITCZ during the LIA.

The 21-year periodicity associated with the solar cycle influences the pattern of tropical rainfall, favoring the displacement and contraction of the ITCZ towards the equator while intensifying the SAMS. On the other hand, the 80-year periodicities associated with the positive phase of the AMO can weaken the intensity of the SAMS. Additionally, periodicities in the range of 2 to 8 years modulate the internal variability of changes in the ITCZ, especially during La Niña events, resulting in an intensification of precipitation in the NEB. Furthermore, the 24-year periodicity related to the PDO and the 64-year periodicity associated with the negative phase of the AMO contribute to the southward displacement of the ITCZ, which in turn increases the intensity of the SACZ.

According to our results, the changes in the position of the ITCZ during the last millennium are influenced by internal forcings such as ENSO, PDO, and AMO, which exhibit spatial patterns of latitudinal displacement between the northern and southern regions. These internal modes of variability can interact and influence the position and intensity of the ITCZ, thereby affecting regional climate variability. According to these observations, during the positive phase of the AMO, we have observed a northward meridional displacement of the ITCZ. On the other hand, during the negative phase of the AMO, a southward displacement.

The results indicate that there is low-frequency variability that affects the distribution of precipitation and has consequences on the intensity/frequency of droughts and floods events in the NEB. These findings suggest that these events are related to the coupling between the ocean and atmosphere. Furthermore, the results of this study have the potential to improve climate models and forecasts by providing a better understanding of the variability of the ITCZ and its impact on extreme precipitation events in northeastern Brazil.

*Author contributions.* Isela Vásquez, Gilvan Sampaio, and Humberto barbosa designed the study. Isela Vásquez conducted the analysis of the models with assistance from Arturo Sanchez and David Pareja. Isela Vásquez wrote the paper, incorporating contributions from all co-authors.

*Competing interests.* The authors declare that they have no conflict of interest.

*Acknowledgements.* This study was supported in part by the Grants FAPESP (Fundacão de Amparo à Pesquisa do Estado de São Paulo) project (Number 2019/08726-7 to Isela Vásquez and 2020/02737-4 to Giselle Utida); Climate Research and Education in the Americas using Tree-ring and Speleothem Examples (PIRE-CREATE) project (Number 2017-50085-3); CNPq (*Conselho Nacional de Desenvolvimento Científico e Tecnológico*) project *Programa de monitoramento da desertificação por satélite no Semiárido brasileiro*, under the Grant/Award Number (403223/2021-0 to H.A.B); RTI-09427-B-C22, Project KUK AHPAN *Amenaza y Riesgo Sísmico en América Central y Sureste de España* (RTI-09427-B-C22 to J. G. Rejas).

**References**

Adam, O., Schneider, T., Brient, F., and Bischoff, T.: Relation of the double-ITCZ bias to the atmospheric energy budget in climate models, Geophysical Research Letters, 43, 7670–7677, 2016.

Adler, R. F., Huffman, G. J., Chang, A., Ferraro, R., Xie, P.-P., Janowiak, J., Rudolf, B., Schneider, U., Curtis, S., Bolvin, D., et al.: The version-2 global precipitation climatology project (GPCP) monthly precipitation analysis (1979–present), Journal of hydrometeorology, 4, 1147–1167, 2003.

Apaéstegui, J., Cruz, F. W., Sifeddine, A., Vuille, M., Espinoza, J., Guyot, J.-L., Khodri, M., Strikis, N., Santos, R., Cheng, H., et al.: Hydroclimate variability of the northwestern Amazon Basin near the Andean foothills of Peru related to the South American Monsoon System during the last 1600 years, Climate of the Past, 10, 1967–1981, 2014.

Asmerom, Y., Baldini, J. U., Prufer, K. M., Polyak, V. J., Ridley, H. E., Aquino, V. V., Baldini, L. M., Breitenbach, S. F., Macpherson, C. G., and Kennett, D. J.: Intertropical convergence zone variability in the Neotropics during the Common Era, Science advances, 6, eaax3644, 2020.

Berry, G. and Reeder, M. J.: Objective identification of the intertropical convergence zone: Climatology and trends from the ERA-Interim, Journal of climate, 27, 1894–1909, 2014.

Bird, B. W., Abbott, M. B., Vuille, M., Rodbell, D. T., Stansell, N. D., and Rosenmeier, M. F.: A 2,300-year-long annually resolved record of the South American summer monsoon from the Peruvian Andes, Proceedings of the National Academy of Sciences, 108, 8583–8588, 2011.

Bischoff, T. and Schneider, T.: Energetic constraints on the position of the intertropical convergence zone, Journal of Climate, 27, 4937–4951, 2014.

Boos, W. R. and Korty, R. L.: Energy budget control of the regional ITCZ: Theory and application to mid-Holocene rainfall, in: AGU Fall Meeting Abstracts, vol. 2016, pp. A53G–05, 2016.

Broccoli, A. J., Dahl, K. A., and Stouffer, R. J.: Response of the ITCZ to Northern Hemisphere cooling, Geophysical Research Letters, 33, 2006.

Buckley, M. W. and Marshall, J.: Observations, inferences, and mechanisms of the Atlantic Meridional Overturning Circulation: A review, Reviews of Geophysics, 54, 5–63, 2016.

Byrne, M. P., Pendergrass, A. G., Rapp, A. D., and Wodzicki, K. R.: Response of the intertropical convergence zone to climate change: Location, width, and strength, Current climate change reports, 4, 355–370, 2018.

Campos, J., Cruz, F., Ambrizzi, T., Deininger, M., Vuille, M., Novello, V., and Strikis, N.: Coherent South American Monsoon variability during the last millennium revealed through high-resolution proxy records, Geophysical Research Letters, 46, 8261–8270, 2019.

Carvalho, L. M. V. d. and Cavalcanti, I. F.: The South American Monsoon System (SAMS), The monsoons and climate change, pp. 121–148, 2016.

Chang, P., Yamagata, T., Schopf, P., Behera, S., Carton, J., Kessler, W., Meyers, G., Qu, T., Schott, F., Shetye, S., et al.: Climate fluctuations of tropical coupled systems—the role of ocean dynamics, Journal of Climate, 19, 5122–5174, 2006.

Chaves, R. R. and Cavalcanti, I. F. A.: Atmospheric circulation features associated with rainfall variability over southern Northeast Brazil, Monthly Weather Review, 129, 2614–2626, 2001.

Chiang, J. C. and Bitz, C. M.: Influence of high latitude ice cover on the marine Intertropical Convergence Zone, Climate Dynamics, 25, 477–496, 2005.

Chiang, J. C., Kushnir, Y., and Giannini, A.: Deconstructing Atlantic Intertropical Convergence Zone variability: Influence of the local cross-equatorial sea surface temperature gradient and remote forcing from the eastern equatorial Pacific, Journal of Geophysical Research: Atmospheres, 107, ACL–3, 2002.

Chikira, M. and Sugiyama, M.: A cumulus parameterization with state-dependent entrainment rate. Part I: Description and sensitivity to temperature and humidity profiles, Journal of the atmospheric sciences, 67, 2171–2193, 2010.

Deser, C., Alexander, M. A., Xie, S.-P., Phillips, A. S., et al.: Sea surface temperature variability: Patterns and mechanisms, Annu. Rev. Mar. Sci, 2, 115–143, 2010.

Donohoe, A., Marshall, J., Ferreira, D., and Mcgee, D.: The relationship between ITCZ location and cross-equatorial atmospheric heat transport: From the seasonal cycle to the Last Glacial Maximum, Journal of Climate, 26, 3597–3618, 2013.

D'Agostino, R., Brown, J. R., Moise, A., Nguyen, H., Silva Dias, P. L., and Jungclaus, J.: Contrasting southern hemisphere monsoon response: MidHolocene orbital forcing versus future greenhouse gas–induced global warming, Journal of Climate, 33, 9595–9613, 2020.

Enfield, D. B. and Mayer, D. A.: Tropical Atlantic sea surface temperature variability and its relation to El Niño-Southern Oscillation, Journal of Geophysical Research: Oceans, 102, 929–945, 1997.

Enfield, D. B., Mestas-Nuñez, A. M., and Trimble, P. J.: The Atlantic multidecadal oscillation and its relation to rainfall and river flows in the continental US, Geophysical Research Letters, 28, 2077–2080, 2001.

Eyring, V., Bony, S., Meehl, G. A., Senior, C. A., Stevens, B., Stouffer, R. J., and Taylor, K. E.: Overview of the Coupled Model Intercomparison Project Phase 6 (CMIP6) experimental design and organization, Geoscientific Model Development, 9, 1937–1958, 2016.

Giannini, A., Chiang, J. C., Cane, M. A., Kushnir, Y., and Seager, R.: The ENSO teleconnection to the tropical Atlantic Ocean: Contributions of the remote and local SSTs to rainfall variability in the tropical Americas, Journal of Climate, 14, 4530–4544, 2001.

Green, B. and Marshall, J.: Coupling of trade winds with ocean circulation damps ITCZ shifts, Journal of Climate, 30, 4395–4411, 2017.

Green, B., Marshall, J., and Donohoe, A.: Twentieth century correlations between extratropical SST variability and ITCZ shifts, Geophysical Research Letters, 44, 9039–9047, 2017.

Grimm, A. M.: The El Niño impact on the summer monsoon in Brazil: regional processes versus remote influences, Journal of Climate, 16, 263–280, 2003.

Grimm, A. M.: How do La Niña events disturb the summer monsoon system in Brazil?, Climate Dynamics, 22, 123–138, 2004.

Grimm, A. M. and Saboia, J. P.: Interdecadal variability of the South American precipitation in the monsoon season, Journal of Climate, 28, 755–775, 2015.

Grimm, A. M. and Tedeschi, R. G.: ENSO and extreme rainfall events in South America, Journal of Climate, 22, 1589–1609, 2009.

Hajima, T., Watanabe, M., Yamamoto, A., Tatebe, H., Noguchi, M., Abe, M., Ohgaito, R., Ito, A., Yamazaki, D., Okajima, H., et al.: Description of the MIROC-ES2L Earth system model and evaluation of its climate–biogeochemical processes and feedbacks, Geoscientific Model Development Discussion, 5, 2019.

Hastenrath, S.: Interannual variability and annual cycle: Mechanisms of circulation and climate in the tropical Atlantic sector, Monthly Weather Review, 112, 1097–1107, 1984.

Hastenrath, S. and Greischar, L.: Circulation mechanisms related to northeast Brazil rainfall anomalies, Journal of Geophysical Research: Atmospheres, 98, 5093–5102, 1993.

Haug, G. H., Hughen, K. A., Sigman, D. M., Peterson, L. C., and Röhl, U.: Southward migration of the intertropical convergence zone through the Holocene, Science, 293, 1304–1308, 2001.

Haug, G. H., Gunther, D., Peterson, L. C., Sigman, D. M., Hughen, K. A., and Aeschlimann, B.: Climate and the collapse of Maya civilization, Science, 299, 1731–1735, 2003.

Henke, L. M., Lambert, F. H., and Charman, D. J.: Was the Little Ice Age more or less El Niño-like than the Medieval Climate Anomaly? Evidence from hydrological and temperature proxy data, Climate of the Past, 13, 267–301, 2017.

Hodrick, R. J. and Prescott, E. C.: Postwar US business cycles: an empirical investigation, Journal of Money, credit, and Banking, pp. 1–16, 1997.

Hwang, Y.-T. and Frierson, D. M.: Link between the double-Intertropical Convergence Zone problem and cloud biases over the Southern Ocean, Proceedings of the National Academy of Sciences, 110, 4935–4940, 2013.

Jones, P. D. and Mann, M. E.: Climate over past millennia, Reviews of Geophysics, 42, 2004.

Kang, S. M., Held, I. M., Frierson, D. M., and Zhao, M.: The response of the ITCZ to extratropical thermal forcing: Idealized slab-ocean experiments with a GCM, Journal of Climate, 21, 3521–3532, 2008.

Kerr, R. A.: A North Atlantic climate pacemaker for the centuries, Science, 288, 1984–1985, 2000.

Kleppe, J., Brothers, D. S., Kent, G. M., Biondi, F., Jensen, S., and Driscoll, N. W.: Duration and severity of Medieval drought in the Lake Tahoe Basin, Quaternary Science Reviews, 30, 3269–3279, 2011.

Knight, J. R., Allan, R. J., Folland, C. K., Vellinga, M., and Mann, M. E.: A signature of persistent natural thermohaline circulation cycles in observed climate, Geophysical Research Letters, 32, 2005.

Knight, J. R., Folland, C. K., and Scaife, A. A.: Climate impacts of the Atlantic multidecadal oscillation, Geophysical Research Letters, 33, 2006.

Lechleitner, F. A., Breitenbach, S. F., Rehfeld, K., Ridley, H. E., Asmerom, Y., Prufer, K. M., Marwan, N., Goswami, B., Kennett, D. J., Aquino, V. V., et al.: Tropical rainfall over the last two millennia: evidence for a low-latitude hydrologic seesaw, Scientific Reports, 7, 1–9, 2017.

Levine, A. F., Frierson, D. M., and McPhaden, M. J.: AMO forcing of multidecadal Pacific ITCZ variability, Journal of Climate, 31, 5749–5764, 2018.

Li, L., Yu, Y., Tang, Y., Lin, P., Xie, J., Song, M., Dong, L., Zhou, T., Liu, L., Wang, L., et al.: The flexible global ocean-atmosphere-land system model grid-point version 3 (FGOALS-g3): description and evaluation, Journal of Advances in Modeling Earth Systems, 12, e2019MS002 012, 2020.

Liebmann, B. and Smith, C. A.: Description of a complete (interpolated) outgoing longwave radiation dataset, Bulletin of the American Meteorological Society, 77, 1275–1277, 1996.

Lin, P., Yu, Z., Lü, J., Ding, M., Hu, A., and Liu, H.: Two regimes of Atlantic multidecadal oscillation: Cross-basin dependent or Atlantic-intrinsic, Science Bulletin, 64, 198–204, 2019.

Lu, Y., Wu, T., Li, Y., and Yang, B.: Mitigation of the double ITCZ syndrome in BCC-CSM2-MR through improving parameterizations of boundary-layer turbulence and shallow convection, Geoscientific Model Development, 14, 5183–5204, 2021.

Mamalakis, A. and Foufoula-Georgiou, E.: A multivariate probabilistic framework for tracking the intertropical convergence zone: Analysis of recent climatology and past trends, Geophysical Research Letters, 45, 13–080, 2018.

Mamalakis, A., Randerson, J. T., Yu, J.-Y., Pritchard, M. S., Magnusdottir, G., Smyth, P., Levine, P. A., Yu, S., and Foufoula-Georgiou, E.: Zonally contrasting shifts of the tropical rain belt in response to climate change, Nature climate change, 11, 143–151, 2021.

Mann, M. and Emanuel, K.: Global warming, the AMO, and North Atlantic tropical cyclones, Eos, 87, 233–244, 2006.

Mann, M. E., Steinman, B. A., Brouillette, D. J., and Miller, S. K.: Multidecadal climate oscillations during the past millennium driven by volcanic forcing, Science, 371, 1014–1019, 2021.

Maravall, A. and Del Rio, A.: Time aggregation and the Hodrick-Prescott filter, 0108, Banco de España, 2001.

Marengo, J. A.: Interdecadal variability and trends of rainfall across the Amazon basin, Theoretical and applied climatology, 78, 79–96, 2004.

Marengo, J. A., Torres, R. R., and Alves, L. M.: Drought in Northeast Brazil—past, present, and future, Theoretical and Applied Climatology, 129, 1189–1200, 2017.

Marshall, J., Donohoe, A., Ferreira, D., and McGee, D.: The ocean's role in setting the mean position of the Inter-Tropical Convergence Zone, Climate Dynamics, 42, 1967–1979, 2014.

Martín-Rey, M., Rodríguez-Fonseca, B., Polo, I., and Kucharski, F.: On the Atlantic–Pacific Niños connection: A multidecadal modulated mode, Climate dynamics, 43, 3163–3178, 2014.

Martín-Rey, M., Rodríguez-Fonseca, B., and Polo, I.: Atlantic opportunities for ENSO prediction, Geophysical Research Letters, 42, 6802–6810, 2015.

McGee, D., Donohoe, A., Marshall, J., and Ferreira, D.: Changes in ITCZ location and cross-equatorial heat transport at the Last Glacial Maximum, Heinrich Stadial 1, and the mid-Holocene, Earth and Planetary Science Letters, 390, 69–79, 2014.

McGee, D., Moreno-Chamarro, E., Green, B., Marshall, J., Galbraith, E., and Bradtmiller, L.: Hemispherically asymmetric trade wind changes as signatures of past ITCZ shifts, Quaternary Science Reviews, 180, 214–228, 2018.

McGregor, S. and Timmermann, A.: The effect of explosive tropical volcanism on ENSO, Journal of climate, 24, 2178–2191, 2011.

Miller, G. H., Geirsdóttir, Á., Zhong, Y., Larsen, D. J., Otto-Bliesner, B. L., Holland, M. M., Bailey, D. A., Refsnider, K. A., Lehman, S. J., Southon, J. R., et al.: Abrupt onset of the Little Ice Age triggered by volcanism and sustained by sea-ice/ocean feedbacks, Geophysical research letters, 39, 2012.

Mitchell, T. P. and Wallace, J. M.: The annual cycle in equatorial convection and sea surface temperature, Journal of Climate, 5, 1140–1156, 1992.

Molion, L. C. B.: Amazonia rainfall and its variability, Hydrology and water manegement in the humid tropics. BONELL, M.; HUFSCHMIDT, MM, pp. 99–111, 1993.

Moreno-Chamarro, E., Marshall, J., and Delworth, T.: Linking ITCZ migrations to the AMOC and North Atlantic/Pacific SST decadal variability, Journal of Climate, 33, 893–905, 2019.

Moura, A. D. and Shukla, J.: On the dynamics of droughts in northeast Brazil: Observations, theory and numerical experiments with a general circulation model, Journal of the atmospheric sciences, 38, 2653–2675, 1981.

Nobre, P. and Shukla, J.: Variations of sea surface temperature, wind stress, and rainfall over the tropical Atlantic and South America, Journal of climate, 9, 2464–2479, 1996.

Novello, V. F., Cruz, F. W., Karmann, I., Burns, S. J., Stríkis, N. M., Vuille, M., Cheng, H., Lawrence Edwards, R., Santos, R. V., Frigo, E., et al.: Multidecadal climate variability in Brazil's Nordeste during the last 3000 years based on speleothem isotope records, Geophysical Research Letters, 39, 2012.

Okumura, Y. and Xie, S.-P.: Interaction of the Atlantic equatorial cold tongue and the African monsoon, Journal of Climate, 17, 3589–3602, 2004.

Ortega, G., Arias, P. A., Villegas, J. C., Marquet, P. A., and Nobre, P.: Present-day and future climate over central and South America according to CMIP5/CMIP6 models, International Journal of Climatology, 41, 6713–6735, 2021.

Oueslati, B. and Bellon, G.: Convective entrainment and large-scale organization of tropical precipitation: Sensitivity of the CNRM-CM5 hierarchy of models, Journal of Climate, 26, 2931–2946, 2013.

Oueslati, B. and Bellon, G.: The double ITCZ bias in CMIP5 models: interaction between SST, large-scale circulation and precipitation, Climate dynamics, 44, 585–607, 2015.

Peterson, L. C. and Haug, G. H.: Variability in the mean latitude of the Atlantic Intertropical Convergence Zone as recorded by riverine input of sediments to the Cariaco Basin (Venezuela), Palaeogeography, Palaeoclimatology, Palaeoecology, 234, 97–113, 2006.

Philander, S., Gu, D., Lambert, G., Li, T., Halpern, D., Lau, N., and Pacanowski, R.: Why the ITCZ is mostly north of the equator, Journal of climate, 9, 2958–2972, 1996.

Philander, S. G.: El Niño, La Niña, and the southern oscillation, International geophysics series, 46, X–289, 1989.

Predybaylo, E., Stenchikov, G. L., Wittenberg, A. T., and Zeng, F.: Impacts of a Pinatubo-size volcanic eruption on ENSO, Journal of Geophysical Research: Atmospheres, 122, 925–947, 2017.

Richter, I. and Tokinaga, H.: An overview of the performance of CMIP6 models in the tropical Atlantic: mean state, variability, and remote impacts, Climate Dynamics, 55, 2579–2601, 2020.

Richter, I. and Xie, S.-P.: On the origin of equatorial Atlantic biases in coupled general circulation models, Climate Dynamics, 31, 587–598, 2008.

Richter, I., Xie, S.-P., Wittenberg, A. T., and Masumoto, Y.: Tropical Atlantic biases and their relation to surface wind stress and terrestrial precipitation, Climate dynamics, 38, 985–1001, 2012.

Richter, I., Xie, S.-P., Behera, S. K., Doi, T., and Masumoto, Y.: Equatorial Atlantic variability and its relation to mean state biases in CMIP5, Climate dynamics, 42, 171–188, 2014.

Samanta, D., Karnauskas, K. B., and Goodkin, N. F.: Tropical Pacific SST and ITCZ biases in climate models: double trouble for future rainfall projections?, Geophysical Research Letters, 46, 2242–2252, 2019.

Schneider, T.: Feedback of atmosphere-ocean coupling on shifts of the Intertropical Convergence Zone, Geophysical Research Letters, 44, 11–644, 2017.

Schneider, T., Bischoff, T., and Haug, G. H.: Migrations and dynamics of the intertropical convergence zone, Nature, 513, 45–53, 2014.

Schulz, M. and Stattegger, K.: SPECTRUM: Spectral analysis of unevenly spaced paleoclimatic time series, Computers & Geosciences, 23, 929–945, 1997.

Seager, R., Naik, N., Baethgen, W., Robertson, A., Kushnir, Y., Nakamura, J., and Jurburg, S.: Tropical oceanic causes of interannual to multidecadal precipitation variability in southeast South America over the past century, Journal of Climate, 23, 5517–5539, 2010.

Servain, J.: Simple climatic indices for the tropical Atlantic Ocean and some applications, Journal of Geophysical Research: Oceans, 96, 15 137–15 146, 1991.

Shonk, J. K., Demissie, T. D., and Toniazzo, T.: A double ITCZ phenomenology of wind errors in the equatorial Atlantic in seasonal forecasts with ECMWF models, Atmospheric Chemistry and Physics, 19, 11 383–11 399, 2019.

Song, F. and Zhang, G. J.: The impacts of horizontal resolution on the seasonally dependent biases of the Northeastern Pacific ITCZ in coupled climate models, Journal of Climate, 33, 941–957, 2020.

Song, X. and Zhang, G. J.: Convection parameterization, tropical Pacific double ITCZ, and upper-ocean biases in the NCAR CCSM3. Part I: Climatology and atmospheric feedback, Journal of Climate, 22, 4299–4315, 2009.

Song, X. and Zhang, G. J.: The roles of convection parameterization in the formation of double ITCZ syndrome in the NCAR CESM: I. Atmospheric processes, Journal of Advances in Modeling Earth Systems, 10, 842–866, 2018.

Stríkis, N. M., Cruz, F. W., Cheng, H., Karmann, I., Edwards, R. L., Vuille, M., Wang, X., de Paula, M. S., Novello, V. F., and Auler, A. S.: Abrupt variations in South American monsoon rainfall during the Holocene based on a speleothem record from central-eastern Brazil, Geology, 39, 1075–1078, 2011.

Stríkis, N. M., Chiessi, C. M., Cruz, F. W., Vuille, M., Cheng, H., de Souza Barreto, E. A., Mollenhauer, G., Kasten, S., Karmann, I., Edwards, R. L., et al.: Timing and structure of Mega-SACZ events during Heinrich Stadial 1, Geophysical Research Letters, 42, 5477–5484A, 2015.

Sulca, J. C. and Rocha, R. P. d.: Influence of the Coupling South Atlantic Convergence Zone-El Niño-Southern Oscillation (SACZ-ENSO) on the Projected Precipitation Changes over the Central Andes, Climate, 9, 77, 2021.

Sun, B., Wang, H., Li, H., Zhou, B., Duan, M., and Li, H.: A Long-Lasting Precipitation Deficit in South China During Autumn-Winter 2020/2021: Combined Effect of ENSO and Arctic Sea Ice, Journal of Geophysical Research: Atmospheres, 127, e2021JD035 584, 2022.

Tedeschi, R. G., Cavalcanti, I. F., and Grimm, A. M.: Influences of two types of ENSO on South American precipitation, International Journal of Climatology, 33, 1382–1400, 2013.

Tian, B. and Dong, X.: The Double-ITCZ Bias in CMIP3, CMIP5, and CMIP6 Models Based on Annual Mean Precipitation, Geophysical Research Letters, 47, 1–20, 2020.

Timmermann, A., Okumura, Y., An, S.-I., Clement, A., Dong, B., Guilyardi, E., Hu, A., Jungclaus, J., Renold, M., Stocker, T. F., et al.: The influence of a weakening of the Atlantic meridional overturning circulation on ENSO, Journal of climate, 20, 4899–4919, 2007.

Tomaziello, A. C. N., Carvalho, L. M., and Gandu, A. W.: Intraseasonal variability of the Atlantic Intertropical Convergence Zone during austral summer and winter, Climate Dynamics, 47, 1717–1733, 2016.

Utida, G., Cruz, F. W., Etourneau, J., Bouloubassi, I., Schefuß, E., Vuille, M., Novello, V. F., Prado, L. F., Sifeddine, A., Klein, V., et al.: Tropical South Atlantic influence on Northeastern Brazil precipitation and ITCZ displacement during the past 2300 years, Scientific reports, 9, 1–8, 2019.

Uvo, C. R. B.: A Zona de Convergência Intertropical (ZCIT) e sua relação com a precipitação da Região Norte do Nordeste Brasileiro, INPE, 1989.

Vásquez, P. I. L., de Araujo, L. M. N., Molion, L. C. B., de Araujo Abdalad, M., Moreira, D. M., Sanchez, A., Barbosa, H. A., and Rotunno Filho, O. C.: Historical analysis of interannual rainfall variability and trends in southeastern Brazil based on observational and remotely sensed data, Climate Dynamics, 50, 801–824, 2018.

Viana, J. C. C., Sifeddine, A., Turcq, B., Albuquerque, A. L. S., Moreira, L. S., Gomes, D. F., and Cordeiro, R. C.: A late Holocene paleoclimate reconstruction from Boqueirão Lake sediments, northeastern Brazil, Palaeogeography, Palaeoclimatology, Palaeoecology, 415, 117–126, 2014.

Vuille, M., Burns, S. J., Taylor, B. L., Cruz, F. W., Bird, B. W., Abbott, M. B., Kanner, L. C., Cheng, H., and Novello, V. F.: A review of the South American monsoon history as recorded in stable isotopic proxies over the past two millennia, Climate of the Past, 8, 1309–1321, 2012.

Waliser, D. and Jiang, X.: Tropical meteorology and climate: Intertropical Convergence Zone, in 'Encyclopedia of Atmospheric Sciences', 2015.

Waliser, D. E. and Gautier, C.: A satellite-derived climatology of the ITCZ, Journal of climate, 6, 2162–2174, 1993.

Wang, X., Edwards, R. L., Auler, A. S., Cheng, H., Kong, X., Wang, Y., Cruz, F. W., Dorale, J. A., and Chiang, H.-W.: Hydroclimate changes across the Amazon lowlands over the past 45,000 years, Nature, 541, 204–207, 2017.

Wills, R. C., Schneider, T., Wallace, J. M., Battisti, D. S., and Hartmann, D. L.: Disentangling global warming, multidecadal variability, and El Niño in Pacific temperatures, Geophysical Research Letters, 45, 2487–2496, 2018.

Wills, R. C., Battisti, D. S., Proistosescu, C., Thompson, L., Hartmann, D. L., and Armour, K. C.: Ocean circulation signatures of North Pacific decadal variability, Geophysical Research Letters, 46, 1690–1701, 2019.

Yin, L., Fu, R., Shevliakova, E., and Dickinson, R. E.: How well can CMIP5 simulate precipitation and its controlling processes over tropical South America?, Climate Dynamics, 41, 3127–3143, 2013.

Yukimoto, S., Kawai, H., Koshiro, T., Oshima, N., Yoshida, K., Urakawa, S., Tsujino, H., Deushi, M., Tanaka, T., Hosaka, M., et al.: The Meteorological Research Institute Earth System Model version 2.0, MRI-ESM2. 0: Description and basic evaluation of the physical component, Journal of the Meteorological Society of Japan. Ser. II, 2019.

Zebiak, S. E.: Air–sea interaction in the equatorial Atlantic region, Journal of Climate, 6, 1567–1586, 1993.

Zermeño-Diaz, D. M. and Zhang, C.: Possible root causes of surface westerly biases over the equatorial Atlantic in global climate models, Journal of climate, 26, 8154–8168, 2013.

Zhang, G. J., Song, X., and Wang, Y.: The double ITCZ syndrome in GCMs: A coupled feedback problem among convection, clouds, atmospheric and ocean circulations, Atmospheric Research, 229, 255–268, 2019.